# Origin of ocean island basalts in the West African passive margin without mantle plume involvement

Iyasu Getachew Belay[1], Ryoji Tanaka [1], Hiroshi Kitagawa[1], Katsura Kobayashi[1] & Eizo Nakamura[1]

The geochemical variabilities in intraplate basalts (IB) from the West African passive margin (WAPM) region, have generally been employed to indicate the presence of recycled materials in an associated upwelling mantle plume. However, the absence of time-progressive linear hotspot tracks in WAPM-IB make it difficult to explain their genesis solely by the mantle plume hypothesis. Here, we show that the Sr–Nd–Hf–Pb isotopic variations in basalts from most of the WAPM-IB could have mainly attributed to the derivation from two types of fusible regions of the refertilized subcontinental lithospheric mantle (SCLM) and the sub-lithospheric mantle. The locations and magma genesis of WAPM-IB are strongly related to the distance from the Mesozoic rift axis and the structure of the rifted SCLM. The melting of the source region can possibly be attributed to small-scale mantle convection at the base of the SCLM without the involvement of a mantle plume.

---

[1] The Pheasant Memorial Laboratory for Geochemistry and Cosmochemistry, Institute for Planetary Materials, Okayama University, Misasa, Tottori 682-0193, Japan. Correspondence and requests for materials should be addressed to R.T. (email: ryoji@misasa.okayama-u.ac.jp)

The genesis of Ocean Island Basalt (OIB) has been widely debated for more than four decades. The generally accepted whole-mantle convection theory is supported by geochemical data on OIBs, which indicates that the source mantle contains various types of recycled materials that were transported through subduction or delamination processes. Recent tomographic images have shown a broad quasi-vertical conduit, which extends from an ultralow-velocity zone at the base of the lower mantle to the source region of major OIB[1]. Time-progressive linear hotspot tracks aligned with plate motion and large igneous provinces where voluminous magmatism occurred within a few million years have been considered as evidence for the deep mantle plume hypothesis. On the other hand, geodynamical studies have proposed both plume and non-plume hypotheses for the initiation of OIB magmatism, e.g. upwelling mantle plumes[2], lithospheric control[3], small-scale convection[4], and asthenospheric shear[5].

The OIB suites, which are located western offshore of the West Africa (the Madeira Islands, the Canary Islands, the Cape Verde Islands, and the oceanic Cameroon volcanic line [CVL]) comprise linear volcanic chains[6] (Fig. 1). Basaltic rocks in these OIB suites and their closely related continental volcanic regions (the Atlas Mountains and the continental CVL), denoted as West African passive margin intraplate basalts (WAPM-IB) in this study, are characterized by a highly alkaline affinity with occurrences of nephelinite and/or carbonatite and prolonged volcanic activity lasting ~20–140 million years[7–12] (Fig. 1). Variations in elemental and isotopic compositions of basaltic rocks from WAPM-IB have been generally explained by the involvement of plume components, involving recycled crustal/lithosphere materials[13]. However, the lack of a clear age progression within volcanic chains for the WAPM-IB[8,9,12,14,15] is inconsistent with the formation of hotspot tracks by melting of stationary plumes.

OIB from the WAPM is unique when compared to other OIB suites because of their association with widely distributed high S-wave velocity (Vs) zones in the shallow upper mantle (Fig. 1). These high Vs zones were interpreted as the remnants of buoyant ancient continental lithosphere, which was fragmented during the opening of the ocean basin[16,17]. Indeed, it has been suggested that geochemical characteristics of WAPM-IB magmas could have been influenced by plume–lithosphere interaction, perhaps involving the entrainment of the delaminated lithospheric mantle into the upwelling mantle plume[18–20]. However, because continental lithosphere consists of a large variety of geological components, so-far proposed recycled components in the mantle plume can be derived from the remnants of continental lithosphere[17]. Thus it is important to evaluate whether the geochemical variations of magmas in the WAPM-IB can be

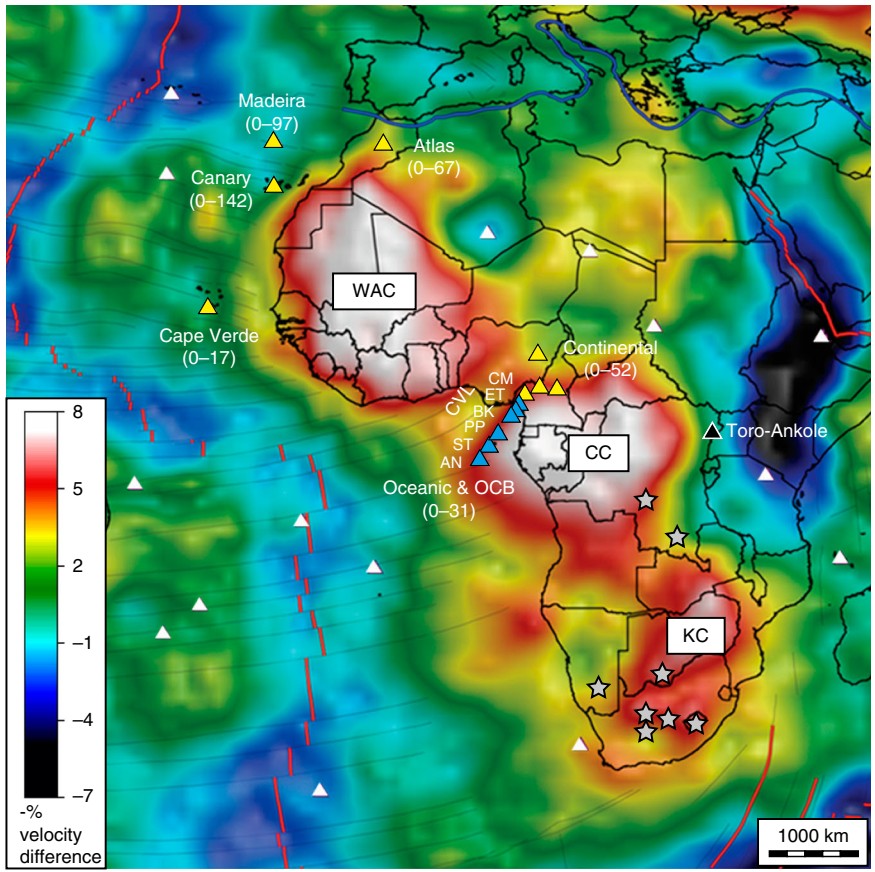

**Fig. 1** Location of West African passive margin intraplate basalts. Light blue triangles show the oceanic and ocean–continental boundary (OCB) Cameroon volcanic line (CVL) volcanoes, which are the focus of this study: AN Annobòn, ST São Tomé, PP Principe, BK Bioko, ET Etínde, CM Mt. Cameroon. Yellow triangles represent the volcanoes whose isotopic compositions were compiled in this study. Numbers shown in a bracket are the eruption ages for each volcanic field in millions of years[7–12]. The age for the Madeira and the Canary islands includes the data for neighbouring seamounts. The black triangle indicates the Toro-Ankole region where the pyroxenite xenolith in the subcontinental lithospheric mantle was obtained. White triangles indicate other volcanoes. WAC West African Craton, CC Congo Craton, KC Kalahari Craton. The grey star represents the location of Group 1 kimberlite that was used for the estimation of the refertilized SCLM compositions. Tomographic model image (Vs models, reference velocity is 4.5 km/s) at depths of 100–175 km for the African continent and Atlantic Ocean is after ref. [17] (reprinted from Lithos, vol. 112, Ultradeep continental roots and their oceanic remnants: A solution to the geochemical "mantle reservoir" problem?, O'Reilly et al. p. 1047, with permission from Elsevier)

explained without a plume component and instead by the non-plume hypothesis proposed by geophysical and geodynamical studies[3,5,21].

Among WAPM-IB, the CVL follows an almost linear trans-lithospheric discontinuity between the Congo and West African cratons[16], and this discontinuity straddles both oceanic and continental regions (Fig. 1). Although the involvement of SCLM in the source of CVL magmas was invoked by many studies for continental and ocean–continental boundary (OCB) volcanoes, it is still controversial whether the magmatic activities of the CVL were triggered by Cenozoic plume activity[1,11,22]. Here we report on major and trace element concentrations and high-precision Sr, Nd, Hf, and Pb isotopic data for 90 basaltic samples from five CVL volcanoes from oceanic and OCB regions: Annobòn, São Tomé, Principe, Bioko, and Etínde (Supplementary Data 1). We also make use of the high-precision isotopic data from Mt. Cameroon lavas previously analysed in our laboratory ($N = 26$)[20]. Then we examine the geochemical variation of the oceanic and OCB CVL to determine whether it can be explained solely by the lithospheric and asthenospheric mantle components that are present beneath the CVL without considering any Cenozoic plume materials. Finally, the non-plume hypothesis for the origin of WAPM-IB is discussed in the context of the geochemical variation of WAPM-IB.

## Results

**Geochemical characteristics of CVL basalts.** Trace element patterns, of all the CVL samples studied here, show general enrichments in highly incompatible elements and large variations in moderately incompatible P, Zr, Hf, and Ti (Fig. 2). In this study, the samples are classified into Types 1 and 2 based on the Hf/Sm and Ti/Gd results. Samples having both $[Hf/Sm]_{PM}$ and $[Ti/Gd]_{PM}$ values of <1 are defined as Type 1, the others are classified as Type 2. The ratio bracketed with $_{PM}$ is the primitive mantle normalized ratio (Supplementary Data 1). Type 1 samples occur in all volcanic regions, whereas Type 2 samples exist only in the SW part of the volcanoes studied here (Annobòn, São Tomé, Principe, and Bioko), except for two samples (1982-1 and 1982-2) from Mt. Cameroon. Although the $[Ti/Gd]_{PM}$ values of these two samples are slightly >1 (1.04 and 1.06), other trace element patterns are indistinguishable from other samples from Mt. Cameroon. Thus all samples from Mt. Cameroon are classified as Type 1.

The above noted negative anomalies of Zr, Hf, and Ti in Type 1 lavas (Fig. 2) are suggestive of a carbonatite or $CO_2$-rich silicate melt metasomatized SCLM origin, under garnet–peridotite stability conditions[23]. Involvement of $CO_2$-rich melt in the source region of the volcanoes studied here was suggested previously for Etínde[24] and São Tomé[25], but it could have been involved in all of the other CVL magmas studied here. Type 1 lavas show further negative anomalies of K and Pb (Fig. 2). Corresponding trace element patterns resemble those of Group 1 kimberlites from both erupted samples and melt preserved in high-Mg high-density fluids in diamond xenocrysts[26] (Fig. 2). On the contrary, the trace element patterns of Group 2 kimberlites show enrichments of U and Ba relative to Nb and no negative Pb anomaly; this is distinct from the trace element patterns of the CVL lavas studied here.

Among the Type 1 samples, distinct Sr, Nd, Hf, and Pb isotopic compositions (Fig. 3) are observed between northeastern (NE; Mt. Cameroon and Etínde) and southwestern (SW; Annobòn, São Tomé, and Principe) regions. The Type 1 NE samples have higher $^{87}Sr/^{86}Sr$, $^{206}Pb/^{204}Pb$, and $^{208}Pb/^{204}Pb$ and lower $^{143}Nd/^{144}Nd$ and $^{176}Hf/^{177}Hf$ relative to Type 1 SW samples. On the plot of $^{206}Pb/^{204}Pb$ vs. $^{207}Pb/^{204}Pb$ diagram, the Type 1 NE samples

show higher $^{206}Pb/^{204}Pb$ at a given $^{207}Pb/^{204}Pb$. Although most of the primitive mantle-normalized incompatible elements in all the Type 1 samples show similar patterns, P and Ba record different patterns when comparing Type 1 NE samples to those of a Type 1 SW samples. Type 1 NE samples show a negative anomaly of P and depletion of Ba relative to Th, but these features are not observed for Type 1 SW samples (Fig. 2). The distinct geochemical behaviours of Type 1 NE and Type 1 SW samples can be observed by plotting the $[P/Nd]_{PM}$ and $[Ba/Th]_{PM}$ (Fig. 4). The $[P/Nd]_{PM}$ and $[Ba/Th]_{PM}$ of the Type 1 NE samples are clearly <1 (Fig. 4). The phosphorous content of the mantle-derived melt is primarily controlled by accessory phosphate minerals[27]. Interestingly, the P/Nd is positively correlated with Ba/Th and K/La. The depletion of P and the positive correlation between P/Nd, Ba/Th, and K/Nd for the NE volcanoes could be attributed to the presence of a complex K–Ba–phosphate phase that is stable in the near-solidus phase at 4–7 GPa[28]. Alternatively, it could represent the remnants of a residual phosphate (e.g. apatite) and K-rich (e.g. K-richterite) phase mixture, which originated from the lithospheric mantle. Thus the source of Type 1 NE lavas should include these metasomatized SCLM[28]. Type 1 samples from Bioko Island, located between the NE and SW regions, show intermediate elemental and isotopic characteristics between those of the Type 1 NE and SW samples (Figs. 3 and 4). Thus it is clear that the source materials of the Type 1 CVL lavas studied here continuously vary from NE to SW regions.

The distinction of Type 1 and 2 samples is evident in their different trace elements and isotopic compositions (Figs. 2–4). Most of the Type 2 samples extend to higher $^{208}Pb/^{204}Pb$ at a given value of $^{206}Pb/^{204}Pb$ and/or lower $^{206}Pb/^{204}Pb$ at a given value of $^{207}Pb/^{204}Pb$ relative to Type 1. On the plot of $^{87}Sr/^{86}Sr$ vs. $^{143}Nd/^{144}Nd$ and $^{143}Nd/^{144}Nd$ vs. $^{176}Hf/^{177}Hf$ diagrams, Type 2 samples either overlap or extend to higher $^{87}Sr/^{86}Sr$ and lower $^{143}Nd/^{144}Nd$ and $^{176}Hf/^{177}Hf$ compared to the Type 1 samples (Fig. 3). The P/Nd and K/La of the Type 2 samples extend to lower and higher values, respectively, than the Type 1 SW samples (Fig. 4). These geochemical signatures indicate that the variation within Type 2 samples may have been controlled by the mixing of the Type 1 SW source material and the enriched endmember components, denoted as the Type 2 enriched component or Type 2 EC. This Type 2 EC is characterized by high $^{87}Sr/^{86}Sr$, low $^{143}Nd/^{144}Nd$ and $^{176}Hf/^{177}Hf$, and relatively low $^{206}Pb/^{204}Pb$ at given $^{207}Pb/^{204}Pb$ and $^{208}Pb/^{204}Pb$ (Fig. 3).

**Source materials of CVL basalts.** The Sr, Nd, Hf, and Pb isotopic compositions of the CVL lavas studied here are distinct from those of the present-day Atlantic depleted mid-ocean ridge basalt (MORB) source mantle (DMM) (Fig. 3, see "Methods" in detail). As the trace element compositions have revealed, the source for Type 1 magmas has likely been influenced by carbonatitic or ultra-low-degree silicate melts, which were presumably derived from the metasomatized SCLM[13,20]. The refertilized SCLM represented by the low-Vs cratonic roots[16,17] could be the most fusible portion of the SCLM and is the most likely candidate for the reservoir of Type 1 magmas (Fig. 5a). Among the available samples, kimberlite has the potential to reveal the nature and composition of the deepest parts of the refertilized SCLM[29]. The location of kimberlites in Africa mostly cluster around the low-Vs zones on the margins of high Vs cratonic roots or near the cratonic margins[29,30]. Chronological and geochemical data of kimberlites in Africa revealed that the isotopic compositions of Group 1 kimberlite could represent the low-Vs metasomatized zones of the SCLM[29]. The similar trace element pattern between Type 1 lavas and Group 1 kimberlite supports the idea of their genetic linkage.

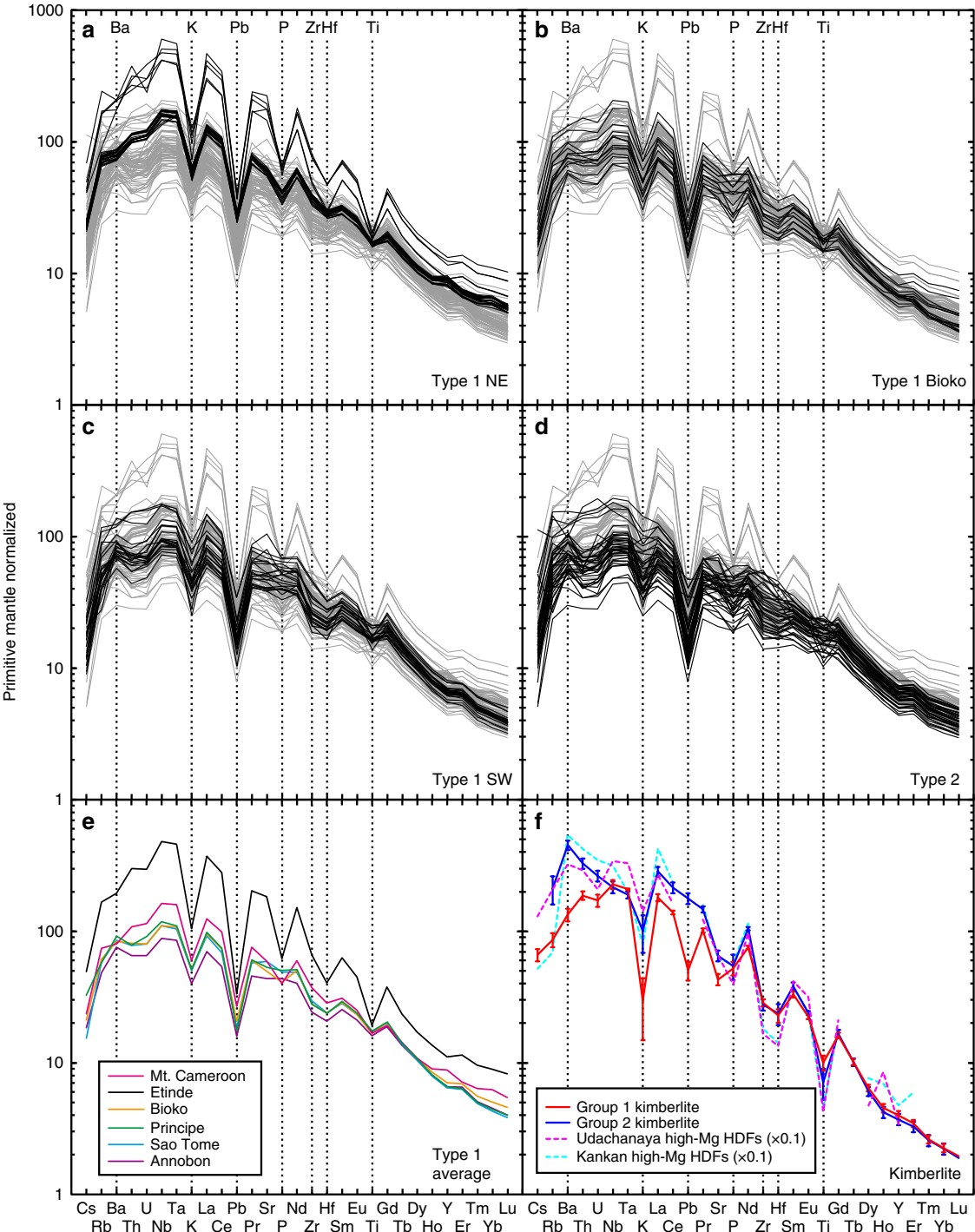

**Fig. 2** Primitive mantle-normalized trace element patterns of Cameroon volcanic line lavas. **a** Type 1 northeastern lavas (Mt. Cameroon[20] and Etínde) shown in black; **b** Type 1 Bioko shown in black; **c** Type 1 southwestern lavas (Principe, São Tomé, and Annobòn) shown in black; **d** Type 2 lavas shown in black; **e** average values of Type 1 lavas for each volcanic location; and **f** average values of Group 1 and Group 2 kimberlite from Africa and high-Mg carbonatitic high-density fluids (HDFs) included in diamond from Udachanaya and Kankan kimberlites[26,67]. Patterns shown in grey in **a–d** are data for all of the Cameroon volcanic line. Error bars of the kimberlite values represent 1 standard error. To facilitate visualization, the data for HDFs were multiplied by 0.1. The altered samples (indicated in Supplementary Data 1) are not plotted. The source data of the compiled values in **f** are provided in ref. [66]. Primitive mantle normalizing values are after ref. [68]. Elements discussed in the main text are labelled with broken lines

Present-day Sr, Nd, Hf, and Pb isotopic compositions of the refertilized SCLM were estimated from the isotopic compositions of African Group1 kimberlite (Fig. 3). The Group 2 kimberlite component is unlikely to be the source of CVL magmas because Group 2 kimberlites were too low in $^{206}Pb/^{204}Pb$, $^{143}Nd/^{144}Nd$ and $^{176}Hf/^{177}Hf$ and too high in $^{87}Sr/^{86}Sr$[31,32] relative to the

samples studied here. Thus the Group 2 kimberlite-source mantle was not considered in this study. The Sr, Nd, Hf, and Pb isotopic compositions of the kimberlite-source mantle were calculated as follows. First, the initial isotopic compositions of kimberlite magma at the time of eruption were calculated. The isotopic compositions of the kimberlite-source mantle for each data set

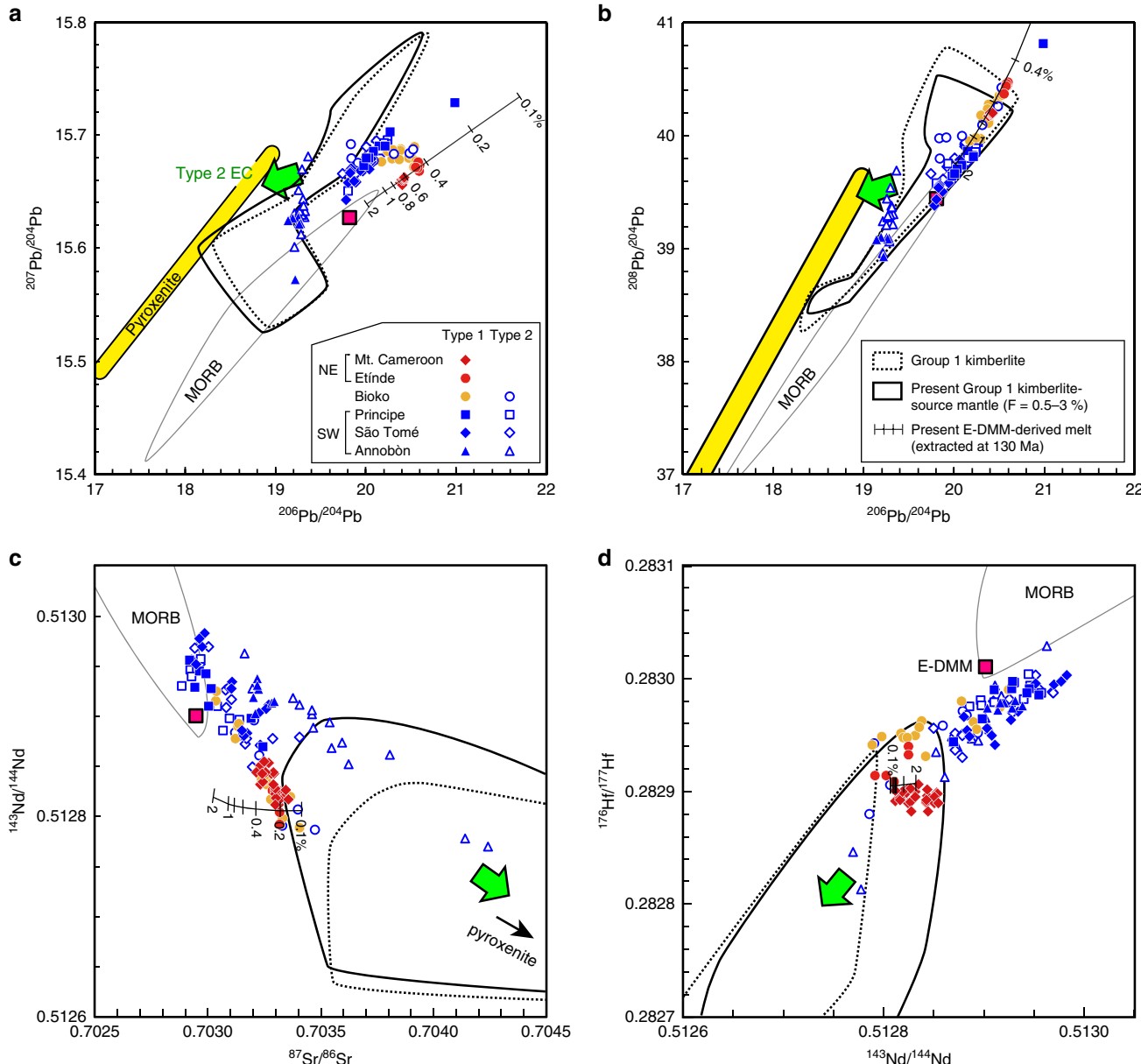

**Fig. 3** Sr–Nd–Hf–Pb isotopic compositions of Cameroon volcanic line lavas. **a** $^{206}Pb/^{204}Pb$ vs. $^{207}Pb/^{204}Pb$, **b** $^{206}Pb/^{204}Pb$ vs. $^{208}Pb/^{204}Pb$, **c** $^{87}Sr/^{86}Sr$ vs. $^{143}Nd/^{144}Nd$, and, **d** $^{143}Nd/^{144}Nd$ vs. $^{176}Hf/^{177}Hf$. The ranges for the isotopic compositions of Group 1 kimberlite and of the calculated Group 1 kimberlite-source subcontinental lithospheric mantle (SCLM) are shown in black broken and solid curves, respectively. The source data of the compiled values for Group 1 kimberlites erupted between 71 and 114 Ma are provided in ref. [66]. The calculated present-day isotopic compositions of the enriched part of the depleted mid-ocean ridge basalt source mantle (E-DMM)-derived melt, which formed during the continental breakup at 130 Ma are shown with black lines (see Supplementary Fig. 1 for further details). The digits in the black lines denote the degree of partial melt at 130 Ma. The yellow region is the pyroxenite vein/layer xenolith derived from the SCLM of the Congo craton[36]. The 2 SD error bars are smaller than the symbols for Cameroon volcanic line data

were assumed to be identical to their calculated initial composition at the time of eruption. Second, the Rb, Sr, Sm, Nd, Lu, Hf, U, Th, and Pb contents of the kimberlite-source mantle were determined using a non-modal batch melting equation (Eq. (1) in "Methods") using a melt composition for each kimberlite composition. The used source mineral mode, partition coefficients, and the $P$ values in Eq. (1) in "Methods" were after ref. [33]. The degree of melting for the production of kimberlite melt is not well constrained, but it should be less than a few percent[33]. Thus the calculation was performed using a fraction of melting of between 0.5 and 3%. Finally, the Sr, Nd, Hf, and Pb isotopic ingrowth in the kimberlite-source mantle to present-day

were calculated using the calculated initial isotopic compositions and parent/daughter element ratios. The calculated result (Fig. 3) demonstrates that the Type 1 SW magma, except for one Principe sample, could have been produced by the mixing of Group 1 kimberlite-source mantle and the DMM. Nevertheless, the $^{207}Pb/^{204}Pb$ data of Type 1 NE samples were distinct from the range of Group 1 kimberlite-source mantle, which also cannot be explained by the mixing of Group 1 kimberlite-source mantle and DMM.

Another possible originator of the refertilized SCLM is asthenospheric mantle-derived melt that metasomatized the SCLM during the rifting that was associated with continental

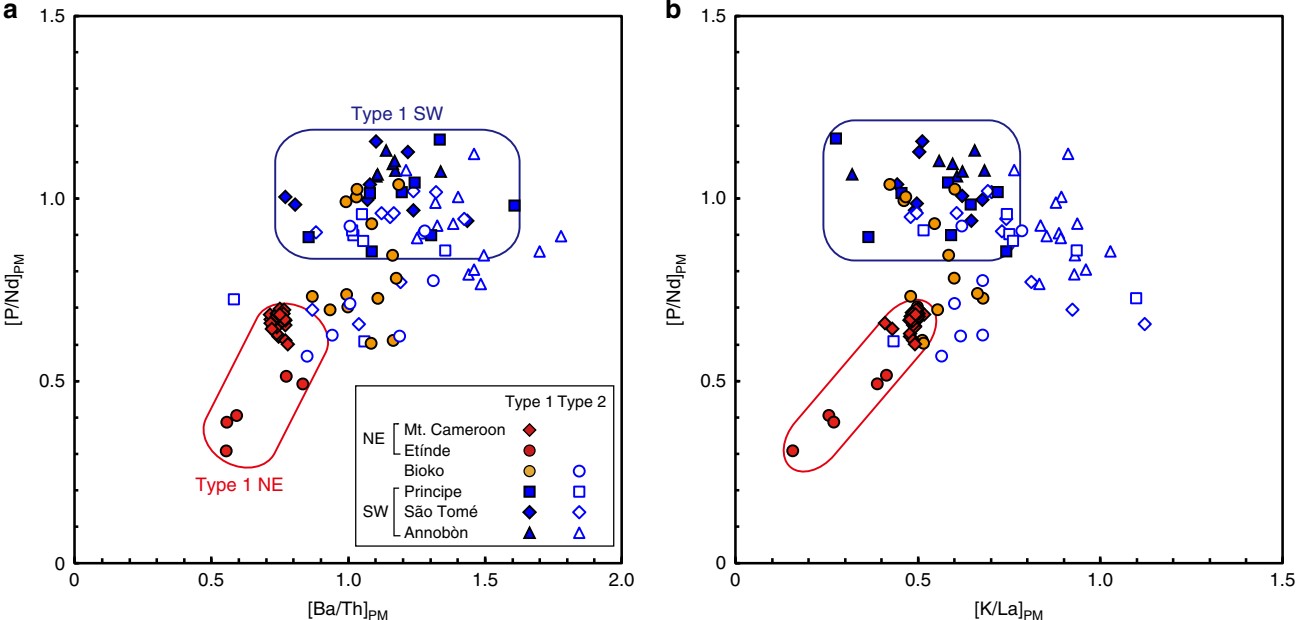

**Fig. 4** Primitive mantle-normalized P/Nd, Ba/Th, and K/La variations of Cameroon volcanic line lavas. **a** [Ba/Th]$_{PM}$ vs. [P/Nd]$_{PM}$ and **b** [K/La]$_{PM}$ vs. [P/Nd]$_{PM}$. Primitive mantle normalizing values are after ref. [68]

breakup[13,20] (Fig. 5b). The isotopic composition of the ancient asthenospheric mantle-derived melt, which metasomatized the lithospheric mantle (Fig. 5b), was calculated based on the elemental and isotopic composition of the present-day DMM with various degrees of melting over various age ranges. The calculation method is described in "Methods" and the calculated result is shown in Supplementary Fig. 1. The results suggest that the Sr, Nd, Hf, and Pb isotopic ranges of Type 1 NE samples can be reproduced by the low-degree (<~0.4%) melting of the enriched part of the DMM (E-DMM) during continental breakup at ~130 Ma (Fig. 3 and Supplementary Fig. 1). Thus this metasomatized SCLM, with isotopic compositions that are largely reflective of the Mesozoic asthenospheric mantle-derived E-DMM-type melt, is the most likely source component of the parental melt of Type 1 NE lavas. However, the Sr, Nd, Hf, and Pb isotopic compositions of Type 1 SW and Type 2 samples cannot be explained by the Mesozoic E-DMM component as a major source material (Fig. 3 and Supplementary Fig. 1).

Most of the Type 2 samples also plotted in the range of the Pb isotopic compositions of the Group 1 kimberlite source, but the identified Type 2 EC suggests the presence of distinct components from the Group 1 kimberlite source (Fig. 3). The isotopic compositions of the Type 2 EC resemble the enriched mantle 1 (EM1) component that could be entrained from the ambient mantle into the upwelling mantle plume[34]. The source of EM1 was estimated as either ancient pelagic sediments in the source or delaminated SCLM in previous studies[34]. However, the actual material that could have been the source of their EM1 components has not been identified. The isotopic trend indicative of Type 2 EC, found in continental CVL lavas along with elevated $^{187}$Os/$^{188}$Os, was interpreted as the involvement of continental crust, although the possibility for the involvement of pyroxenite could not be rejected[35]. No correlation between Nb/U and Pb isotopic compositions for the Type 2 CVL samples studied here (figure is not shown) indicates that the continental crust is an unlikely source for the Type 2 EC. Thus the possible candidate for the source of Type 2 EC was investigated by compiling all the published Sr–Nd–Pb isotopic data of mantle xenoliths from cratonic and near-cratonic regions of Africa from the GEOROC

database (Supplementary Fig. 2). Among the compiled data sets, only the pyroxenite vein/layer in the SCLM peridotite of the Congo–Tanzania craton, collected from the Toro-Ankole volcanic region, SW Uganda[36] (Fig. 1) has a Sr, Nd, and Pb isotopic composition (no Hf isotopic data were reported) consistent with the source of Type 2 EC (Supplementary Fig. 2). Any other mantle xenolith does not fit the assumed Type 2 EC, in particular for the $^{206}$Pb/$^{204}$Pb vs. $^{208}$Pb/$^{204}$Pb diagram. Thus the pyroxenite vein/layer in the cratonic SCLM is the best candidate for the Type 2 EC. The pyroxenite vein/layer was interpreted as forming between ~0.2 and ~1 Ga and having coexisted with the highly metasomatized SCLM[36]. Thus it is likely that the source of Type 2 magmas could coexist with the source region of SW part of the CVL as indicated by our results.

Therefore, the isotope systematics of the CVL samples studied here can be explained by a mixture of the three refertilized SCLM components (Group 1 kimberlite-source mantle, Mesozoic E-DMM-derived melt, and pyroxenite) and the DMM. Thus no external Cenozoic plume component was necessary for the genesis of CVL magmas.

## Discussion

The compiled Sr, Nd, Hf, and Pb isotopic compositions of the WAPM-IB are plotted with those of CVL samples (Fig. 6). Also, the principal component analysis using the three Pb isotopic compositions of the WAPM-IBs were performed (Fig. 7). The first and second principal component plots in Fig. 7 indicates that the major sources of the Canary Islands, Atlas Mountains, continental CVL, and most of the Cape Verde Islands could be the same four identified components as those of the CVL data studied. Exceptionally, the data from the Madeira Islands show distinct compositions from the range of the CVL in Fig. 7. The distinct compositions for Madeira were also observed for Sr, Nd, and Hf isotopic compositions (Fig. 6). Thus the source of Madeira could involve the undefined source materials. The $^{176}$Hf/$^{177}$Hf values for the southern Cape Verde islands samples are higher than the kimberlite-source mantle at a given $^{143}$Nd/$^{144}$Nd. These isotopic compositions were interpreted as possibly originating from part of the upwelling mantle plume or lithospheric

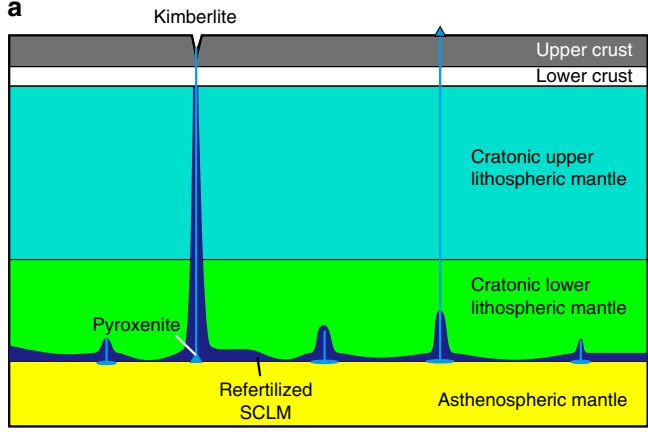

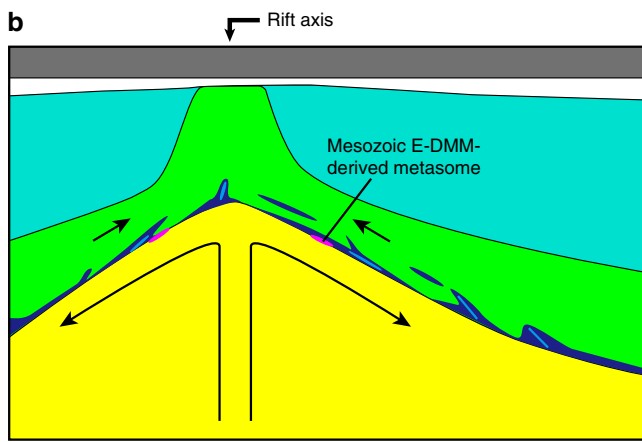

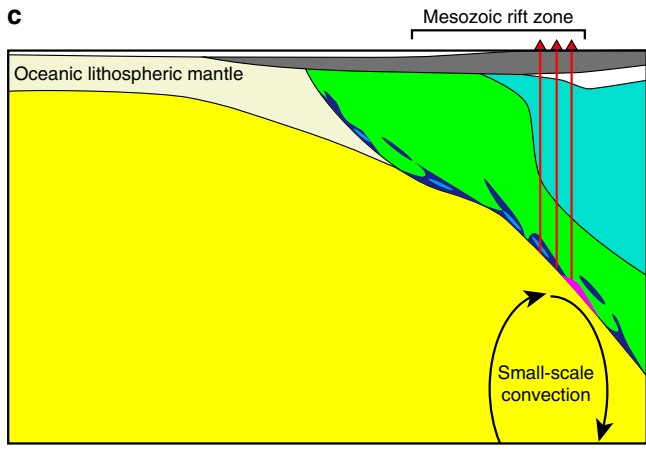

**Fig. 5** Schematic cross-sections illustrating the source materials and the associated melting process of Cameroon volcanic line magmas. **a** Craton before the continental breakup. Image of the subcontinental lithospheric mantle (SCLM) after ref. [69], which demonstrates how the lowermost region of the cratonic lower lithospheric mantle could have refertilized (refertilized SCLM, shown in blue). **b** During the initial stage of Mesozoic, continental breakup based on the depth-dependent extension model after ref. [45]. The low-degree partial melt formed by the upwelling asthenospheric mantle at the rift axis could have metasomatized the lower part of the lithospheric mantle (Mesozoic enriched part of the depleted mid-ocean ridge basalt source mantle-derived melt, shown in magenta). **c** Present. Several tens of million years after the initial rifting, the small-scale convection (SSC) could be developed beneath the step of SCLM[4,57]. The SSC may transform heat to the base of the SCLM, facilitating melting of refertilized SCLM. Melt derived from the pyroxenite vein or layer that formed in the cratonic SCLM (pyroxenite in light blue) during the Archaean or Proterozoic was also involved in the SCLM-derived melt

the compiled WAPM-IB results, CVL NE samples show the highest $^{206}Pb/^{204}Pb$ values except for one anomalous Principe sample (Fig. 6). In other WAPM-IB regions, only some of the Middle Atlas basanites show similar Pb isotopic compositions to those of the CVL NE samples (Figs. 6 and 7). The distinct Pb isotopic values of these high $^{206}Pb/^{204}Pb$ NE CVL and Atlas lavas from the typical HIMU components, represented by St. Helena basalts (Supplementary Fig. 1), indicates that the involvement of St. Helena-type plume is unlikely for the generation of CVL and Atlas magmas as suggested previously[13]. The Pb isotopic trend of these high $^{206}Pb/^{204}Pb$ for the Atlas samples consistently extend to the basanite lavas in the Canary Islands (up to $^{206}Pb/^{204}Pb =$ 20.27)[40,41]. A common characteristic of these high $^{206}Pb/^{204}Pb$ CVL and Atlas samples (>20.1) is their silica-deficient composition (basanite or nephelinite)[39–42]. During the Late Triassic and Early Jurassic, the Moroccan microcontinent separated from the north-western African continent, forming the Atlas Rift (Fig. 8). This was followed by rift structure inversion during the Cenozoic, which formed the current Atlas Mountains[43]. The SCLM beneath the Atlas Mountains were metasomatized by the asthenospheric mantle-derived carbonatitic melt at <200 Ma during the continental breakup[44]. The carbonatite melt-metasomatized clinopyroxene separates, found in the SCLM peridotite xenoliths from the Quaternary Middle Atlas lavas, consistently demonstrate a similar Pb isotopic trend to that of CVL NE magmas[44] (Supplementary Fig. 2). Consequently, these high $^{206}Pb/^{204}Pb$ isotopic trend is a unique characteristic of the Mesozoic E-DMM-derived melt component, resulting from a low-degree melt during the Jurassic–Cretaceous continental breakup, and as such no Cenozoic plume activity was necessary to form the high $^{238}U/^{204}Pb$ signature in the WAPM-IB.

The high potential for the refertilized SCLM as a major magma source of WAPM-IB was proposed by seismic tomographic images[16,17] and numerical models[45]. These studies revealed that remnants of cratonic lithospheric fragments were widespread beneath WAPM-IB regions. The three-dimensional high-Vs zone beneath the Congo and West African cratons and the Atlantic Ocean, which represents refractory cores of the SCLM, has an irregular shape at the lithosphere–asthenosphere boundary[16]. The irregular shape of the three-dimensional high-Vs zone indicates the presence of metasomatized lower-Vs refertilized parts of the SCLM beneath and/or around the refractory SCLM[16]. Furthermore, the refertilized parts of the lithosphere can create lithosphere-scale pathways for fluids and heat to be transported around the SCLM[16]. For example, the P- and S-wave tomographic images beneath the CVL show tabular-shaped

mantle[37]. Extraction of melt from the garnet-bearing mantle rapidly evolves the $^{176}Hf/^{177}Hf$ in the residual mantle and therefore may not allow accurate prediction of $^{176}Hf/^{177}Hf$ of the present mantle composition[32]. Thus deviation of $^{176}Hf/^{177}Hf$ for southern Cape Verde islands samples can be caused by the inaccurate estimation of the kimberlite-source mantle.

The involvement of remnant African SCLM fragments in the source of WAPM-IBs has been proposed by geochemical studies[34,38–41]. However, these previous studies argued that the genesis of WAPM-IB was mainly attributable to the upwelling mantle plume, which contained recycled HIMU (high-μ = elevated $^{238}U/^{204}Pb$) as an intrinsic plume component, and the SCLM fragments were minor components trapped in the plume or interacted with the plume-derived melt. Among all

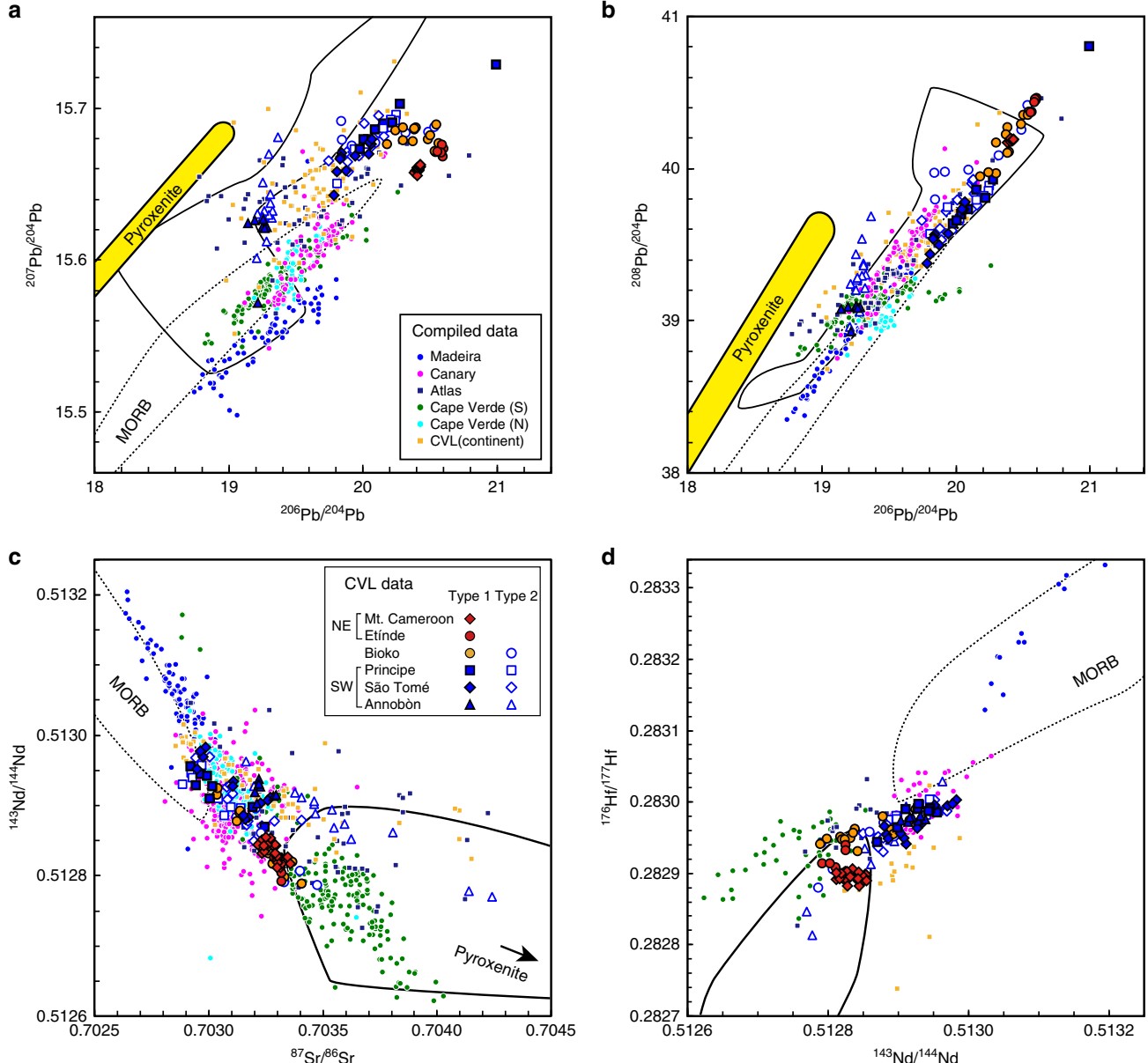

**Fig. 6** Sr–Nd–Hf–Pb isotopic compositions of Cameroon volcanic line (CVL) basalts plotted with other West African passive margin intraplate basalts. **a** $^{206}Pb/^{204}Pb$ vs. $^{207}Pb/^{204}Pb$; **b** $^{206}Pb/^{204}Pb$ vs. $^{208}Pb/^{204}Pb$; **c** $^{87}Sr/^{86}Sr$ vs. $^{143}Nd/^{144}Nd$; and **d** $^{143}Nd/^{144}Nd$ vs. $^{176}Hf/^{177}Hf$. The data for Mt. Cameroon are from ref. [20]. The range for mid-ocean ridge basalt (MORB) shown with broken curves was compiled with high-precision Atlantic MORB data collected between 30°N and 30°S; these data were analysed either by the double spike or Tl-addition methods and were compiled from the PetDB database (www.earthchem.org/petdb). The data for the Madeira Islands, the Canary Islands, the Atlas Mountains, the southern and northern Cape Verde islands, and the continental CVL were compiled by the GEOROC database. All the Pb isotopic data for the Canary Islands, Cape Verde, and the continental sector of the CVL were analysed either by the double spike or Tl-addition methods. Because no Pb isotopic data analysed by either the double spike or Tl-addition methods were available for the Madeira and Atlas samples, the Pb data analysed by conventional methods were compiled for samples from these regions. The solid curve is the calculated Kimberlite-source mantle composition. The 2 SD error bars are smaller than the symbols for CVL data. The source data of the compiled values are provided in ref. [66].

low-velocity zones that extend to a depth of ~300 km[46]. A multimode inversion of surface- and S-wave tomographic images beneath the Canary Island and Atlas regions show broad low-velocity zones[47]. The sharp seismic velocity difference along the SCLM of the Congo and West African cratonic margin, where the Canary Islands and CVL are located, cannot be explained by the temperature difference alone and requires a compositional gradient[16]. Beneath Cape Verde, there is a sharp contrast between the low and high P- and S-wave seismic velocity anomalies along the SE islands[48], which could arise from the sharp contrast in

lithospheric thickness observed for this region[16,17]. Owing to the detached depleted cratonic SCLM being buoyant relative to the convecting mantle, it is likely that these SCLM materials became widespread beneath the Atlantic Ocean after its opening[49]. The refertilized Archaean–Proterozoic SCLM-derived mantle xenolith that was partially metasomatized by kimberlitic melt was actually found in the Cape Verde lavas[49,50]. Therefore, the irregular shapes of high-velocity zones beneath the WAPM-IB regions should illustrate the distribution of refertilized zones that underlie the refractory SCLM[16]. This geophysical and

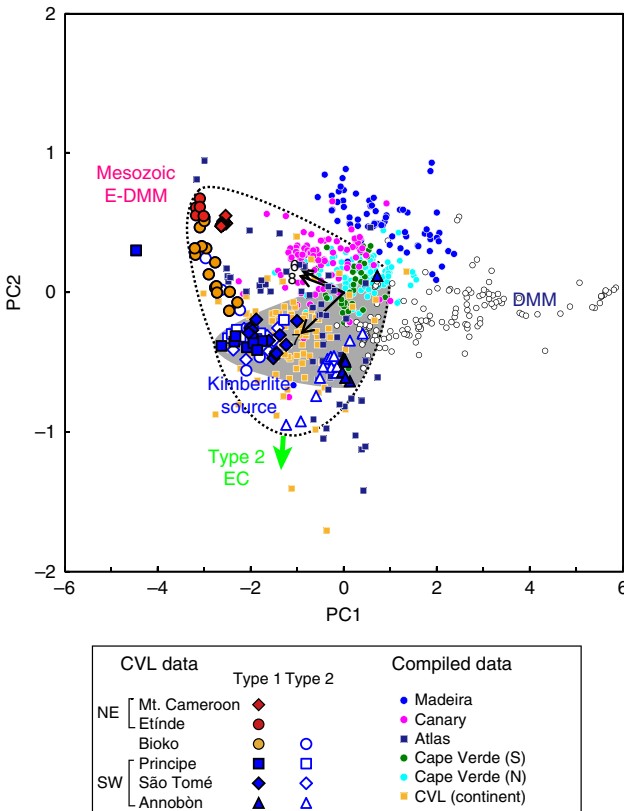

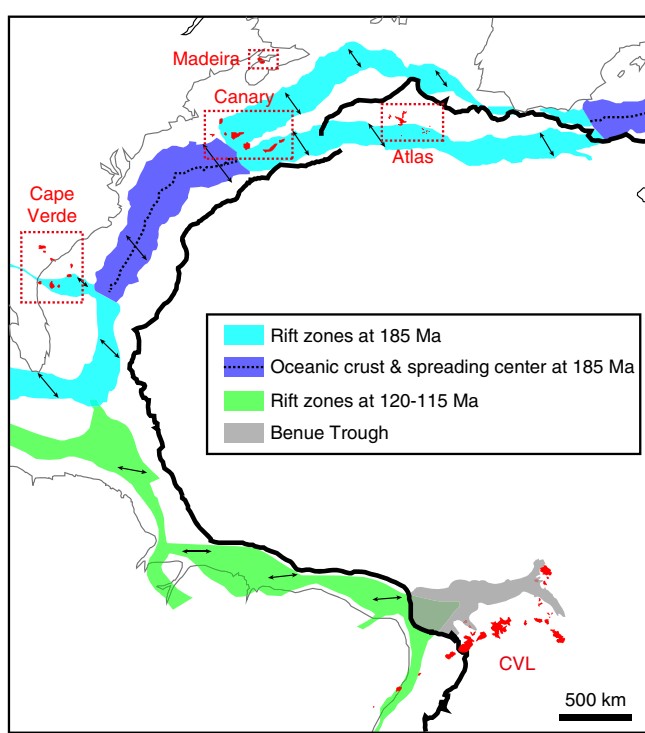

**Fig. 7** Diagram of the principal component (PC) analysis of Pb isotope data. Plot of the first and second PCs (PC1 and PC2) for three Pb isotopic ratios of WAPM-IB and Atlantic MORB samples ($N = 707$). PC1 and PC2 accounted for 94.3 and 4.6 % of the total variability of the data, respectively. The regions shown with broken curves and grey areas are the ranges for the Cameroon volcanic line (CVL) and Type 1 SW lavas, respectively. DMM depleted mid-ocean ridge basalt source mantle, E-DMM enriched part of the DMM. Pb isotopic compositions of the compiled data from the GEOROC database were normalized to $^{206}Pb/^{204}Pb = 16.9424$, $^{207}Pb/^{204}Pb = 15.5003$, and $^{208}Pb/^{204}Pb = 36.7266$ for SRM 981. All the Pb isotopic data applied to the calculations for the PC analysis were analysed by the double spike and Tl-doped methods, except for the Madeira and Atlas data (because all the available Pb data for Madeira and Atlas were analysed by conventional methods). The 2 SD error bars are smaller than the symbols for CVL data. The source data of the compiled values are provided in ref. [66]

**Fig. 8** The reconstructed Mesozoic rift zone of the African and South American continents and Atlantic Ocean and the locations of the West African passive margin intraplate basalts (WAPM-IB). The locations for the WAPM-IB (red) are illustrated with the current distance from the coastline of the African continent (thick black line). The rift zones are at 185 Ma[43] and 120–115 Ma[70]. The Benue Trough, the NW-SW rift depression initiated during the latest Jurassic to Early Cretaceous, are also shown

petrological evidences are consistent with our geochemical result, which reveals the ubiquitous presence of the fertile portion at the lowermost part of the SCLM beneath most of the WAPM-IB region.

Our results infer that the geochemical characteristics of WAPM-IBs is strongly related to the location of the rift axis during the Mesozoic continental breakup and the distribution of the SCLM beneath the Atlantic Ocean. We constructed a map of the African–South American continents and the rift axis during the Jurassic to Cretaceous periods, using data from previous studies (Fig. 8). The map shows that the current position of the Canary Islands, southern area of the Atlas Mountains, southern Cape Verde islands, and the CVL in relation to the coast line of the African continent were located on or near the Mesozoic rift axis (Fig. 8). The location of the Mesozoic rift axis in the Atlantic Ocean should roughly mirror the current position of continent–oceanic boundary. Beneath such a past rift zone, it is likely that the edge of cratonic SCLM, thinned lithospheric

mantle, or underplated lower SCLM[45] present (Fig. 5). The limited occurrence of lavas with high $^{206}Pb/^{204}Pb$ values (up to 20.6) in WAPM-IB at the off-axis inside of the cratonic region (CVL NE and Atlas) supports the notion that off-axis asthenospheric mantle-derived melt, formed at a low-degree of melting and at relatively high pressure (Fig. 5b), were dominated by the most fusible and therefore geochemically enriched source component (i.e. E-DMM)[51]. On the other hand, the locations of Madeira and the northern Cape Verde islands were off the Mesozoic rift axis toward the present Atlantic ridge axis. Although the isotopic data for Madeira and northern Cape Verde islands partially overlap by the range of CVL, most of them show within or around the MORB values (Figs. 6 and 7). Previous studies have suggested that the source of magmas of the northern Cape Verde islands are dominant in the DMM component and less so in SCLM components relative to those of the southern Cape Verde islands[19,38]. Furthermore, $^{3}He/^{4}He$ in olivine phenocrysts in basalts from the northern Cape Verde islands (from 7.0 to 15.7 $R_A$ and mean = 10.4 ± 5.5 $R_A$; $R_A$ is the atmospheric ratio = $1.348 \times 10^{-6}$; error is 2 SD)[19,52], ranging from that in Atlantic MORB glass (8.4 ± 1.4 $R_A$; the data between 30°N and 30°S were downloaded from PetDB database [www.earthchem.org/petdb]) towards the high $^{3}He/^{4}He$ value, indicate the involvement of the relatively primordial mantle component in these magmas[19]. On the contrary, the $^{3}He/^{4}He$ in olivine phenocrysts from other WAPM-IB lavas (no He isotopic data were reported for basalts from the Madeira Islands): from 5.2 to 7.9 $R_A$ and mean = 6.3 ± 1.7 $R_A$ for CVL[53,54], from 5.7 to 9.7 and mean = 7.6 ± 1.5 $R_A$ for Canary Islands[40,55,56], and from 6.3 to 8.9 and mean = 8.1 ± 1.1 $R_A$ for southern Cape Verde islands[19]; range from MORB values extending to lower values,

consistently revealing the involvement of refertilized SCLM components in their source[19,40,54,56].

The generation of refertilized SCLM-derived magmas can be triggered by low-amplitude thermal or compositional anomalies of the sub-lithospheric asthenospheric mantle. Sub-lithospheric small-scale convection (SSC) is the most likely dynamical process of the upper mantle that can initiate melting processes (Fig. 5c). Melting can arise due to small amplitude variations in lithospheric thicknesses or density heterogeneity in the asthenosphere, by steps between continental–oceanic boundary, SCLM blobs, thinned lithosphere, or even due to the instability (e.g. density inversion) of the thickened thermal boundary layer[21,57]. SSC causes asthenosphere of a normal temperature to rise in places of thin lithosphere and sink in zones of thick lithosphere such as beneath the CVL[21]. The upwelled SSC erodes the lithospheric mantle, which is replaced by asthenospheric mantle[4]. A tabular-shaped low-Vs zone is observed beneath the WAPM and could be formed by this process[46], and melting can occur by decompression of asthenospheric mantle and/or by heating of fertile material imbedded in SCLM. Consequently, we conclude that, from a geochemical, geophysical, and geodynamical point of view, genesis of WAPM-IB is strongly controlled by the former location of the Mesozoic rift axis, and a plume rising from the lower mantle is not necessary for the generation of the magmas in these regions.

## Methods

**Samples and analytical methods.** Ninety mafic samples collected from the Annobòn [21], São Tomé [26], Principe [17], Bioko [21], and Etínde [5] volcanoes (the number of samples from each region are shown in brackets) were used in this study. Sample location and petrography were reported elsewhere[54,58]. All experiments were carried out at the PML, IPM, Okayama University[59].

For the bulk rock chemical analyses, the rock specimens were crushed by a jaw crusher to coarse chips of 3–5 mm in diameter, and then chips without weathered parts were carefully hand-picked. Subsequently, the chips were rinsed several times with deionized water in an ultrasonic bath until the supernatants of the water were clear. The chips were then dried in an oven at 100 °C for >8 h and pulverized into fine powder using an alumina ceramic puck mill.

Whole-rock major element compositions, including Cr and Ni contents, were determined by X-ray fluorescence spectrometry (Philips PW 2400) using fused glass. The $H_2O^+$ and FeO contents were determined by gravimetric and titration methods, respectively. Analytical errors (1 SD) for major element analysis were <1%. Trace element compositions were determined by using a Q-pole-type inductively coupled plasma mass spectrometer (Agilent 7500 CS) following methods described elsewhere[60,61]. Typical analytical errors of <5% were obtained for all the analyses of trace elements. All the major and trace element analyses were duplicated and their average values were used.

Isotopic compositions for Sr, Nd, and Pb were determined by thermal ionization mass spectrometry (Thermo TRITON) in static multi collection mode, following methods described elsewhere[59,62]. The Pb isotope composition was determined by double spike method. Hafnium isotopes were measured by a multiple-collector inductively coupled plasma mass spectrometer (Thermo NEPTUNE)[63]. Fractionation of isotope ratios were normalized to $^{86}Sr/^{88}Sr$ = 0.1194, $^{146}Nd/^{144}Nd$ = 0.7219, and $^{179}Hf/^{177}Hf$ = 0.7325. Before decomposing the powdered samples for Sr–Nd and Pb isotopes measurement, the powdered samples were leached using 6 M HCl for 5 and 9 h, respectively, to remove potential contaminants. Total procedural blanks for Sr, Nd, Hf, and Pb were <260, <7, <4, and <20 pg, respectively. The intermediate precision (2 SD) of standard materials during the analyses were: $^{87}Sr/^{86}Sr$ = 0.710279 ± 0.000018 for NIST987 ($N$ = 15) and 0.703705 ± 0.000011 for JB-2 ($N$ = 5), $^{143}Nd/^{144}Nd$ = 0.511749 ± 0.000017 for PML-Nd ($N$ = 17), which corresponds to 0.511885 for La Jolla and 0.513124 ± 0.000017 ($N$ = 7) for JB2, $^{176}Hf/^{177}Hf$ = 0.282178 ± 0.000005 ($N$ = 72) for JMC 14374 and 0.283229 ± 0.000007 ($N$ = 14) for JB3, $^{206}Pb/^{204}Pb$ = 16.9430 ± 0.0042, $^{207}Pb/^{204}Pb$ = 15.5003 ± 0.0045, and $^{208}Pb/^{204}Pb$ 36.730 ± 0.013 for NBS981 ($N$ = 34), and $^{206}Pb/^{204}Pb$ = 18.2975 ± 0.0020, $^{207}Pb/^{204}Pb$ = 15.5393 ± 0.0021, and $^{208}Pb/^{204}Pb$ = 38.2570 ± 0.0047 for JB-3 ($N$ = 2).

**Isotopic compositions of the ancient asthenospheric mantle-derived melt.** The Sr, Nd, Hf, and Pb isotopic compositions of the asthenospheric mantle-derived melt that could have been extracted and metasomatized at various ages were calculated as follows. First, the isotopic compositions of the DMM and the E-DMM at

various ages were calculated from the present asthenospheric mantle compositions. The Sr, Nd, Hf, and Pb isotopic compositions of the present asthenospheric mantle were calculated from data compiled for Atlantic MORB between 30°N and 30°S from the PetDB database (www.earthchem.org/petdb).

Present isotopic compositions of the DMM were determined by averaging the compiled values ($^{87}Sr/^{86}Sr$ = 0.70255, $^{143}Nd/^{144}Nd$ = 0.51313, $^{176}Hf/^{177}Hf$ = 0.28320, $^{206}Pb/^{204}Pb$ = 18.72, $^{207}Pb/^{204}Pb$ = 15.54, and $^{208}Pb/^{204}Pb$ = 38.28) and of E-DMM at the enriched end of the compiled values ($^{87}Sr/^{86}Sr$ = 0.70295, $^{143}Nd/^{144}Nd$ = 0.51290, $^{176}Hf/^{177}Hf$ = 0.28305, $^{206}Pb/^{204}Pb$ = 19.80, $^{207}Pb/^{204}Pb$ = 15.63, and $^{208}Pb/^{204}Pb$ = 39.50). The concentration of parent and daughter elements (Rb, Sr, Sm, Nd, Lu, Hf, U, Th, and Pb) of the DMM and E-DMM were obtained from ref. [64]. The calculation was performed for various ages, namely, 600, 280, 130, and 30 Ma, which correspond to the Pan African cycle, the Artinskian stage when the shallow shelf was developed along the current coast line of West Africa, the opening of the south Atlantic, and the beginning of CVL volcanism, respectively. Second, parent and daughter elemental compositions of the DMM- and E-DMM-derived melt with various degree of melting ($C_L$) were calculated by a non-modal batch melting equation:

$$C_L = \frac{C_0}{D_0 + F(1-P)} \qquad (1)$$

where $C_0$ is the initial concentration in the source of DMM and E-DMM[64], $D_0$ is the distribution coefficient at the start of melting, and $F$ is the fraction of melting. $P$ is defined as $\sum p^i K_0^i$, where $p^i$ and $K_0^i$ are the fractional contributions of the phase to the melt and partition coefficients of phase $i$, respectively. The calculation was performed under garnet–peridotite conditions, because the thickness of the cratonic lithosphere was estimated to be ~200 km and the $^{238}U$–$^{230}Th$ data of Mt. Cameroon lavas revealed the presence of garnet in the source[20]. The following initial modal composition of source peridotite and melting proportion values were used for 17.7 wt.% melt-extracted garnet peridotite at 7 GPa[65]: olivine:clinopyroxene: garnet = 0.58:0.26:0.16 and 0.25:0.51:0.24 for initial mode and melting proportions, respectively. Partition coefficient between mineral (olivine:clinopyroxene:garnet) and silicate melt are $(2 \times 10^{-4}:0.011:2 \times 10^{-4})$ for Rb, $(1 \times 10^{-5}:0.1283:7 \times 10^{-3})$ for Sr, $(7 \times 10^{-5}:0.1873:0.057)$ for Nd, $(7 \times 10^{-4}:0.291:0.217)$ for Sm, $(0.03:0.433:9)$ for Lu, $(4 \times 10^{-3}:0.256:0.5)$ for Hf, $(1.8 \times 10^{-5}:3 \times 10^{-3}:5 \times 10^{-3})$ for U, $(1.2 \times 10^{-5}:1 \times 10^{-3}:1 \times 10^{-3})$ for Th, and $(0:0.072:5 \times 10^{-4})$ for Pb. Decay constants used for the calculations were $1.42 \times 10^{-11}$ for $^{87}Rb$, $6.54 \times 10^{-12}$ for $^{147}Sm$, $1.93 \times 10^{-11}$ for $^{176}Lu$, $1.55125 \times 10^{-10}$ for $^{238}U$, $9.8485 \times 10^{-10}$ for $^{235}U$, and $4.9475 \times 10^{-11}$ for $^{232}Th$. Finally, the current isotopic compositions for the DMM- and E-DMM-derived melt with various degrees of melting are age-corrected, and the result is shown in Supplementary Fig. 1. The source data of the used parameters and calculated values shown in Supplementary Fig. 1 are provided in ref. [66].

## Data availability

All data generated or analysed during this study are included in the supplementary information files and in the cited reference.

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

## Acknowledgements

We thank F.T. Aka for suggesting the project, providing the samples, and discussions. We acknowledge X. Shen, C. Sakaguchi, I. Pineda-Velasco, and K. Tanaka for laboratory assistance; T. Ota, T. Kunihiro, G.E. Bebout, H. Cheng, C. Potiszil, and T.O. Rooney for helpful discussions; and S.Y. O'Reilly for providing the original figure file. Constructive review by G. Ito and anonymous reviewers are highly appreciated. I.G.B was supported by Japanese Government Scholarships and grants for MEXT Program for Education and Research; R.T. was supported by JSPS KAKENHI Grant Number 16K05578; and E.N. was supported by grants for JSPS Asia–Africa Science Platform Program.

## Author contributions

E.N. conceived the project. R.T., H.K. and K.K. designed the project. I.G.B. analysed all the trace elements and isotope data. H.K. determined the K.-Ar age. I.G.B. and R.T. wrote the paper with contributions and edits from all other authors.

## Additional information

**Competing interests:** The authors declare no competing interests.

