## [Peer Review File · Nature Communications]

Reviewers' comments:

Reviewer #1 (Remarks to the Author):

The origin of the Cameroon Line is one of geology's enigmas. There is no compelling explanation. It certainly cannot be explained by the mantle plume hypothesis. The paper by Belay et al has much to commend it. I find no fault with the geochemical arguments and the geochemistry group in Misasa has a reputation for producing data of the highest quality. However I do have concerns about the dynamics. Mantle melting away from plumes and subduction zones is driven by decompression. In their cartoon the authors show an edge driven convection cell at the edge of the SCML and argue that heat conducted from this cell will heat their proposed melting zone at the base on the lithosphere. There are several problems with their proposed model:

- (i) Temperature differences within the asthenosphere are small and they are relying on these temperature differences to provide their required heat. Replacing asthenosphere with asthenosphere is not going to make much difference.
- (ii) The lateral moving asthenosphere will always be separated from the SCLM by a sheared boundary layer, which will minimize its impact on the SCML.
- (iii) As the authors point out the Cameroon produces volcanism over an extended period of time. Once the low melting point in the SCML has melted it cannot melt again and the remaining SCML becomes refractory.
- (iv) Viscous drag from edge driven convection will tend to pull the SCML down and inhibit melting.
- (v) Their model predicts that melting will follow the edge of the SCML. It does not explain the linear trend of the Cameroon Line away from the edge of the SCML.

Although I sympathise with the authors, and have no better explanation for the Cameroon Line, I cannot recommend publication. The Cameroon Line remains an enigma!

Some minor points

Line 11: The term hot spot implies mantle with an excess temperature (above normal) and this type of mantle is only found in plumes. The term hot spot should not be used in this context.

Line 38: Instantaneously is not a few million years!

Line 89-90: Geochemistry can be used to identify the characteristics of the mantle that melted to form a given basalt but not its position in the mantle -plume, SCLM etc or the mechanism of melting (see Hawkesworth and Schersten, 2007 Chem Geol 241, 319-331

Lines 91-103: There is an act of faith here that kimberlites come from the SCML. The SCML is the coldest most refractory part of the mantle. Why should it melt? Even the low melting component can only melt if it decompresses. Kimberlites in South Africa show ages progression, which is used by diamond exploration companies to predict where to find kimberlites indicating that from by melting in ascending mantle plumes. The mixing they demonstrate could equally come from two components in the asthenosphere.

Line 154: insist is the wrong word here. Omit "We thus insist that" and start the sentence with 'Therefore the high-'

Reviewer #2 (Remarks to the Author):

This is a very elegant study that uses a robust, extensive database of new high-quality geochemical analyses to probe the nature and possible origin of the basalts along the Cameroon Line as part of the west African passive margin intraplate basalts. The conclusions are that these basalts show characteristics that do not require a deep-source mantle plume. They have characteristics consistent with the involvement of partial melting of relatively low-melting fractions of metasomatised mantle domains, providing significant support for the growing new conceptual framework that much of the geochemical variation in intraplate basaltic composition that has given rise to the basalt alphabet. This excellent study is a key piece of knowledge to help fill in this giant and complex 4D puzzle.

In addition, the case for a laterally convecting asthenospheric model for heat transfer is very well presented and plausible. The last paragraph, lines 176 – 193 is clear, cogent and brings high-quality geochemical evidence as new and robust knowledge to support this.

This result adds serious weight to the growing acceptance of the concept that Archean lithospheric mantle is buoyant and has persisted in the lithosphere since it formed. It may be metasomatised many times, and such events refertilise the original depleted composition (a process that also

changes the seismic properties). This process makes metasomatised regions denser and of with higher V_p and V_s so that such domains can be recognised in seismic tomography images. Such original and refertilised domains of Archean heritage thus form the dominant lateral and vertical extent of lithospheric mantle today (see Griffin et al., *J Petrology* 2009, 50, 1185-1204, for a more detailed discussion).

Some specific comments to improve clarity of the ideas and conclusions:

1. Lines 21-22: It would be very helpful if there was a little more clarity in how the actual conclusions are expressed. The sentence from line 19 to line 22 is a bit ambiguous, suggesting that the Cameroon line basalts actually derive from melting of metasomatised lithospheric mantle. This is not feasible on mass balance considering the huge erupted volumes of basalt. However, from most of the rest of the manuscript, I infer that the authors really mean to conclude that the magma types with different geochemical signatures (that they so convincingly describe), may be derived from interaction of asthenospheric melts from discrete subcontinental lithospheric mantle (SCLM) previously metasomatised by different metasomatic events. This interaction adds trace elements and distinctive isotopic signatures by supplying small-volume melts from the lower-melting point metasomatised parts.

This just requires some rewording for clarification.

2. Line 53: Note that the line of the Cameroon volcanic follows an almost linear trans-lithospheric discontinuity between two major cratons as described in the reference #27 cited here, and shown in the tomographic image of Supplementary figure 1 (adapted from reference #28). Old (contiguous) lithospheric mantle actually underlies significant parts of the oceanic basin up to 1,000 km west of the "continental boundary" of Africa. Fig 5 in Reference #28 also shows this phenomenon reinforced in the global magnetic data. Such westward extension of the lithospheric mantle supports the concept that interaction of melts with different lithospheric mantle domains determined the trace-element and isotopic characteristics of magmas that ascended in this region, and that interacted with old variously metasomatised lithosphere. See cartoon in reference #28, Figure 9.

3) Lines 87-88: Please see the shapes of the high-velocity volumes of the old refractory lithospheric domains in Figure 10 of reference #27. These represent refractory cores that are irregularly shaped and illustrate that the metasomatised lower-velocity refertilised parts of the SCLM create lithosphere-scale pathways for fluids (and heat) to be transported around old SCLM and along conduits that are variably metasomatised – a very nice additional piece of evidence for the conclusions of this study.

4) Lines 127-128: See comment 2) above.

5) Line 159: The idea here, as well as the Supplementary Figure 1, is after Figure 4 in reference #28, that reference #28 should be noted in the text here. Part of the Cape Verde Islands are actually above a detached “buoyant blob” of SCLM stranded on opening of the ocean basin (see figure 10 from reference #28). And the Re-Os data for SCLM xenoliths from Cape Verde (in reference #30) track the timing of likely metasomatic events.

6) I upload pages 1-4 showing some very minor editorial corrections

In summary:

This is an important contribution to understanding geodynamic processes that shaped the current continental array. The science is based on excellent new geochemical data meticulously interpreted, and a wide understanding of relevant previous work in the the the manuscript scientifically exciting and would like to discuss it with the authors!

Not only is the science exciting and well-founded, the manuscript is also very well written and organised.

It could be published as it stands, but maybe be a little polishing considering the comments above, is in order.

Reviewer #3 (Remarks to the Author):

This manuscript reports on the findings and interpretations of an extensive new data set of trace element and isotope compositions found in the west African passive margin intraplate basalts. The main findings are that most of the lavas of the Cameroon line (CVL) have Pb isotope compositions that fall within those computed for present-day African kimberlites, which the author's associate with subcontinental lithospheric mantle (SCLM). The compositions from Canary and Atlas Mountains span a comparable range of $^{143}\text{Nd}/^{144}\text{Nd}$ vs $^{206}\text{Pb}/^{204}\text{Pb}$ as the CVL. The Madeira and Cape Verde

lavas show compositions that are similar to the CVL compositions, but that also then extend well away from the CVL compositions in Nd-Pb isotope space. The main conclusion is that all of the above WAPM-IB lavas were produced by melting a combination of ambient mantle and SCLM, and none of the compositions require the presence of mantle plumes. This conclusion has large significance to the understanding of hotspot-type volcanic chains, however I am not fully convinced. My main concerns are listed below. I include specific comments and questions as annotations in the manuscript PDF file.

- I. First, the writing was hard for me to follow in several places. This is an aspect that must be improved. Please see my various comments as annotations in the PDF file.
- II. The discussions of the calculations were insufficient for me to understand what the calculations were doing. First and foremost, there was a calculation to arrive at the present day lead isotope composition of SCLM. I do not understand what it is that needs to be calculated? Why cannot the author simply use the measured compositions of the kimberlites? Since the major conclusions hinge on the similarity of the compositions of the WAPM-IB to the SCLM compositions, it is extremely important that the reader understand how the SCLM compositions are defined and what are the likely uncertainties in the calculations. These aspects were not explained.

The other calculations appear to have been of (i) melting of the group 1 kimberlites source and (ii) the melting of the enriched part of the depleted MORB mantle. I did not recognize the result of the former calculation (i) in any of the figures. For both melting calculations, I do not understand how melting changes the isotope composition of the lavas. Melting a homogeneous mantle source does not fractionate isotopes of the same elements from each other; I must therefore infer that the melting calculations assumed the source was isotopically heterogeneous. This must be explained. The processes that control how the isotopic signatures changes with melting must also be explained.

- III. I am most convinced that the CVL lavas were not formed by a plume. I agree that the CVL lavas all fall within the solid green curves in Fig. 1, which I take at face value to represent the African kimberlites compositions. This is good evidence that these lavas do not need a plume source (that is assuming the African kimberlites were not generated as a result of a plume beneath Africa). The lack of an age progression along the CVL is also convincing evidence against a plume origin. Finally the alignment of the CVL with the NE edge of the Congo craton puts it in a favorable orientation to be explained by shallow edge driven convection. These findings are extremely important to a general understanding of the diverse causes of volcanism on earth.

There is also a good evidence that the Canary Islands and Atlas Mountains share the same $^{143}\text{Nd}/^{144}\text{Nd}$ and $^{206}\text{Pb}/^{204}\text{Pb}$ source as the CVL. I would be more convinced if the authors showed the compositions of these other WAPM-IB lavas in $^{207}\text{Pb}/^{204}\text{Pb}$ vs $^{206}\text{Pb}/^{204}\text{Pb}$ and $^{208}\text{Pb}/^{204}\text{Pb}$ vs $^{206}\text{Pb}/^{204}\text{Pb}$ space. I don't understand why they were not. Madeira and Cape Verde have compositions that overlap with, but also extend away from those of the CLV lavas. Thus I don't think it is possible to reject a plume hypothesis for Madeira and Cape Verde. In fact, it may be easier to explain these features in terms of a plume mixing with material from the SCLM. Regarding all of the

above WAPM-IB lavas (Canaries included), the relatively young volcanism on old lithosphere far from any active makes a non-plume origin all the more difficult to explain.

Response to referees

Reviewer #1 (Remarks to the Author):

The origin of the Cameroon Line is one of geology's enigmas. There is no compelling explanation. It certainly cannot be explained by the mantle plume hypothesis. The paper by Belay et al has much to commend it. I find no fault with the geochemical arguments and the geochemistry group in Misasa has a reputation for producing data of the highest quality. However I do have concerns about the dynamics. Mantle melting away from plumes and subduction zones is driven by decompression. In their cartoon the authors show an edge driven convection cell at the edge of the SCML and argue that heat conducted from this cell will heat their proposed melting zone at the base on the lithosphere. There are several problems with their proposed model:

(i) Temperature differences within the asthenosphere are small and they are relying on these temperature differences to provide their required heat. Replacing asthenosphere with asthenosphere is not going to make much difference.

Reply: The explanation for melting of the SCLM by the interaction of asthenospheric mantle with the SCLM is deleted. The revised manuscript emphasized that the melting of the fertile SCLM is attributed by edge-driven small-scale convection as previous geophysical studies explained as a heat source of some WAPM-IB localities. In addition, significant elevation of the temperature is not necessary to melt the refertilized SCLM as shown in Supplementary Fig. 3. These explanations are mainly described in the final paragraphs.

(ii) The lateral moving asthenosphere will always be separated from the SCLM by a sheared boundary layer, which will minimize its impact on the SCML.

Reply: As replied in the previous comment, we emphasized the edge-driven convection for the heat source. The lateral convection, however, can induce the occurrence of edge-driven convection and elevation of temperature both in the lithospheric and asthenospheric mantle in the passive margin. The structure of the lithospheric mantle during continental rifting depends on the structure of lithospheric mantle before rifting. We follow the depth-dependent extension model which proposed the exhumation of large tracts of continental mantle beneath the passive margin.

(iii) As the authors point out the Cameroon produces volcanism over an extended period of time. Once the low melting point in the SCML has melted it cannot melt again and the remaining SCML becomes refractory.

Reply: We agree that the residue can be more refractory after extraction of large degree of melt. However, refractoriness depends on the temperature and degree of melting. Also, exhumation of deep lithospheric mantle during rifting can provide fusible region at the same location. Thus it is possible to generate the metasomatized SCLM-derived low-degree of melt from the same location as we revised.

(iv) Viscous drag from edge driven convection will tend to pull the SCML down and inhibit melting.

Reply: As the studies of Huisman and Beaumont (2011, 2014) presented, the lower part of SCLM can be exhumated during continental rifting. Geophysical data reveals that the deep region of lithospheric mantle beneath the WAPM are eroded, possibly by melting of refertilized SCLM. All these explanation was shown in the last two paragraphs of the revised paper.

(v) Their model predicts that melting will follow the edge of the SCML. It does not explain the linear trend of the Cameroon Line away from the edge of the SCML.

Reply: Figure 1 indicates that the CVL are located along the edge of SCLM.

Although I sympathise with the authors, and have no better explanation for the Cameroon Line, I cannot recommend publication. The Cameroon Line remains an enigma!

Reply: We believe that we have logically explained the magma genesis of CVL using the high-precision geochemical data combined with previously reported geophysical data. To make clear the reviewer's comment, we have revised the manuscript.

Some minor points

Line 11: The term hot spot implies mantle with an excess temperature (above normal) and this type of mantle is only found in plumes. The term hot spot should not be used in this context.

Reply: The term "hotspot" was changed to "Ocean Island ballast (OIB)". However, the term "hotspot tracks" is remained because this term is generally used.

Line 38: Instantaneously is not a few million years!

Reply: "instantaneously" was deleted.

Line 89-90: Geochemistry can be used to identify the characteristics of the mantle that melted to form a given basalt but not its position in the mantle -plume, SCLM etc or the mechanism of melting (see Hawkesworth and Schersten, 2007 Chem Geol 241, 319-331

Reply: This sentence has revised, then moved. After the explanation of possibility of SCLM for their source, the trace element data was used as a supportive data.

Lines 91-103: There is an act of faith here that kimberlites come from the SCML. The SCML is the coldest most refractor part of the mantle. Why should it melt? Even the low melting component can only melt if it decompresses. Kimberlites in South Africa show ages progression, which is used by diamond exploration companies to predict where to find kimberlites indicating that from by melting in ascending mantle plumes. The mixing they demonstrate could equally come from two components in the asthenosphere.

Reply: The text was not revised but adding one reference (Griffin et al., 2009). We agree that the SCLM should be refractory depleted in magmatophile elements. However, here we explained as "refertilized" SCLM where the metasomatic agent was added into the base of the SCLM, thus it can be feasible. The source of the metasomatic agent can be a melt that derived from the upwelling mantle plume as explained in the referred paper (Griffin et al., 2009; 2013). However, we are not arguing if the kimberlite is plume origin or not in this paper.

Line 154: insist is the wrong word here. Omit "We thus insist that" and start the sentence with "Therefore the high-"

Reply: revised.

Reviewer #2 (Remarks to the Author):

This is a very elegant study that uses a robust, extensive database of new high-quality geochemical analyses to probe the nature and possible origin of the basalts along the Cameroon Line as part of the west African passive margin intraplate basalts. The conclusions are that these basalts show characteristics that do not require a deep-source mantle plume. They have characteristics consistent with the involvement of partial melting of relatively low-melting fractions of metasomatised mantle domains, providing significant support for the growing new conceptual framework that much of the geochemical variation in intraplate basaltic composition that has given rise to the basalt alphabet. This excellent study is a key piece of knowledge to help fill in this giant and complex 4D puzzle.

In addition, the case for a laterally convecting asthenospheric model for heat transfer is very well presented and plausible. The last paragraph, lines 176 – 193 is clear, cogent and brings high-quality geochemical evidence as new and robust knowledge to support this.

This result adds serious weight to the growing acceptance of the concept that Archean lithospheric mantle is buoyant and has persisted in the lithosphere since it formed. It may be metasomatised many times, and such events refertilise the original depleted composition (a process that also changes the seismic properties). This process makes metasomatised regions denser and of with higher V_p and V_s so that such domains can be recognised in seismic tomography images. Such original and refertilised domains of Archean heritage thus form the dominant lateral and vertical extent of lithospheric mantle today (see Griffin et al., *J Petrology* 2009, 50, 1185-1204, for a more detailed discussion).

Reply: We are thankful to the reviewer for the positive comment. We revised the manuscript to emphasize these important points.

Some specific comments to improve clarity of the ideas and conclusions:

1. **Lines 21-22:** It would be very helpful if there was a little more clarity in how the actual conclusions are expressed. The sentence from **line 19 to line 22** is ambiguous, suggesting that the Cameroon line basalts actually derive from melting of metasomatised lithospheric mantle. This is not feasible on mass balance considering the huge erupted volumes of basalt. However, from most of the rest of the manuscript, I infer that the authors really mean to conclude that the magma types with different geochemical signatures (that they so convincingly describe), may be derived from interaction of asthenospheric melts from discrete subcontinental lithospheric mantle (SCLM) previously metasomatised by different metasomatic events. This interaction adds trace elements and distinctive isotopic signatures by supplying small-volume melts from the lower-melting point metasomatised parts.

Reply: Since ~40 % of the abstract had to be reduced to fit the journal's requirement, we could not revise the abstract to explain more in detail. In the initial manuscript, the possibility for the involvement of the asthenospheric mantle-derived melt into the parental magma of WAPM-IBs was not clearly described. We understand that the isotopic compositions of the Sr, Nd, and Pb could be significantly influenced by the chemically enriched components derived from the metasomatized SCLM even asthenospheric mantle-derived depleted melt was mixed. However, to generate the asthenospheric mantle-derived melt at deeper level beneath the SCLM, significant thermal anomaly is necessary (Supplementary Fig. 3), which could not be evidenced in this study. Thus, we deleted the ambiguous description which may have suggested the involvement of asthenospheric mantle-derived melt.

This just requires some rewording for clarification.

2. **Line 53:** Note that the line of the Cameroon volcanic follows an almost linear translithospheric discontinuity between two major cratons as described in the reference #27 cited here, and shown in the tomographic image of Supplementary figure 1 (adapted from reference #28). Old (contiguous) lithospheric mantle actually underlies significant parts of the oceanic basin up to 1,000 km west of the “continental boundary” of Africa. Fig 5 in Reference #28 also shows this phenomenon reinforced in the global magnetic data. Such westward extension of the lithospheric mantle supports the concept that interaction of melts with different lithospheric mantle domains determined the trace-element and isotopic characteristics of magmas that ascended in this region, and that interacted with old variously metasomatised lithosphere. See cartoon in reference #28, Figure 9.

Reply: We added following sentences by citing the references in the introduction section.

“Old lithospheric mantle underlies significant parts of the oceanic basin up to 1000 km west of the continental boundary of Africa¹⁶. Such westward extension of the lithospheric mantle beneath the WAPM region indicates that interaction of melts with different lithospheric mantle domains may have determined the geochemical characteristics of magmas that ascended in this region¹⁶.”

“Among the WAPM-IB, the CVL shows unique features in that it follows an almost linear translithospheric discontinuity between Congo and West African cratons¹⁹ that saddles both the oceanic and continental regions (Fig. 1).”

3) **Lines 87-88:** Please see the shapes of the high-velocity volumes of the old refractory lithospheric domains in Figure 10 of reference #27. These represent refractory cores that are irregularly shaped and illustrate that the metasomatised lower-velocity refertilised parts of the SCLM create lithosphere-scale pathways for fluids (and heat) to be transported around old SCLM and along conduits that are variably metasomatised – a very nice additional piece of evidence for the conclusions of this study.

Reply: We agree. The more detailed explanation is added by following this comment.

“The three-dimensional high-velocity zone beneath the Congo craton that represents refractory cores of SCLM shows irregular shape, indicating the metasomatized lower-velocity refertilized parts of the SCLM beneath and/or around the refractory SCLM¹⁹. These refertilized parts can create lithosphere-scale pathways for fluids and heat to be transported around the SCLM¹⁹.”

4) **Lines 127-128:** See comment 2) above.

Reply: We agree. One following sentence was added to explain the seismic evidence for the presence of refertilized portion of SCLM.

“The irregular shape of high-velocity zones in these regions illustrates the distribution of refertilized zones underlie refractory SCLM¹⁹.”

5) **Line 159:** The idea here, as well as the Supplementary Figure 1, is after Figure 4 in reference #28, that reference #28 should be noted in the text here. Part of the Cape Verde Islands are actually above a detached “buoyant blob” of SCLM stranded on opening of the ocean basin (see figure 10 from reference #28). And the Re-Os data for SCLM xenoliths from Cape Verde (in reference #30) track the timing of likely metasomatic events.

Reply: We agree. Revised as:

“Tomographic data revealed that the CVL and Canary Islands are located along or near the edge of the SCLM and part of the Cape Verde are above the a detached buoyant blob of SCLM stranded on opening of the ocean basin¹⁶ (Fig. 1).”

6) I upload pages 1-4 showing some very minor editorial corrections

Reply: All the grammatical and spelling errors were revised.

In summary:

This is an important contribution to understanding geodynamic processes that shaped the current continental array. The science is based on excellent new geochemical data meticulously interpreted, and a wide understanding of relevant previous work in the manuscript scientifically exciting and would like to discuss it with the authors!

Not only is the science exciting and well-founded, the manuscript is also very well written and organised.

It could be published as it stands, but maybe be a little polishing considering the comments above, is in order.

Reviewer #3 (Remarks to the Author):

This manuscript reports on the findings and interpretations of an extensive new data set of trace element and isotope compositions found in the west African passive margin intraplate basalts. The main findings are that most of the lavas of the Cameroon line (CVL) have Pb isotope compositions that fall within those computed for present-day African kimberlites, which the author's associate with subcontinental lithospheric mantle (SCLM). The compositions from Canary and Atlas Mountains span a comparable range of $^{143}\text{Nd}/^{144}\text{Nd}$ vs $^{206}\text{Pb}/^{204}\text{Pb}$ as the CVL. The Madeira and Cape Verde lavas show compositions that are similar to the CVL compositions, but that also then extend well away from the CVL compositions in Nd-Pb isotope space. The main conclusion is that all of the above WAPM-IB lavas were produced by melting a combination of ambient mantle and SCLM, and none of the compositions require the presence of mantle plumes. This conclusion has large significance to the understanding of hotspot-type volcanic chains, however I am not fully convinced. My main concerns are listed below. I include specific comments and questions as annotations in the manuscript PDF file.

I. First, the writing was hard for me to follow in several places. This is an aspect that must be improved. Please see my various comments as annotations in the PDF file.

Reply: We replied for all the annotations as will be explained below.

II. The discussions of the calculations were insufficient for me to understand what the calculations were doing. First and foremost, there was a calculation to arrive at the present day lead isotope composition of SCLM. I do not understand what it is that needs to be calculated? Why cannot the author simply use the measured compositions of the kimberlites? Since the major conclusions hinge on the similarity of the compositions of the WAPM-IB to the SCLM compositions, it is extremely important that the reader understand how the SCLM compositions are defined and what are the likely uncertainties in the calculations. These aspects were not explained.

Reply: Because U/Pb and Th/Pb of the erupted kimberlite magma and of the metasomatic agent that refertilized the SCLM can be different, we first calculated the initial isotopic compositions of the kimberlites. Then, the possible ranges of isotopic compositions of the current refertilized SCLM were calculated using the plausible U/Pb and Th/Pb ratios as explained in the initial and revised manuscripts.

The other calculations appear to have been of (i) melting of the group 1 kimberlites source and (ii) the melting of the enriched part of the depleted MORB mantle. I did not recognize the result of the former calculation (i) in any of the figures. For both melting calculations, I do not understand how melting changes the isotope composition of the lavas. Melting a homogeneous mantle source does not fractionate isotopes of the same elements from each other; I must therefore infer that the melting calculations assumed the source was isotopically heterogeneous. This must be explained. The processes that control how the isotopic signatures changes with melting must also be explained.

Reply: The present isotopic composition depends on the parent/daughter ratio and age after melting and/or metasomatism. Parent/daughter ratio can change with different degree of melting, thus the present isotopic composition can be controlled by the degree of melting. The text and caption of Fig. 5 were revised to make it clear it.

III. I am most convinced that the CVL lavas were not formed by a plume. I agree that the CVL lavas all fall within the solid green curves in Fig. 1, which I take at face value to represent the African kimberlites compositions. This is good evidence that these lavas do not need a plume source (that is assuming the African kimberlites were not generated as a result of a plume beneath Africa). The lack of an age progression along the CVL is also convincing evidence against a plume origin. Finally the alignment of the CVL with the NE edge of the Congo craton puts it in a favorable orientation to be explained by shallow edge driven convection. These findings are extremely important to a general understanding of the diverse causes of volcanism on earth.

There is also a good evidence that the Canary Islands and Atlas Mountains share the same $^{143}\text{Nd}/^{144}\text{Nd}$ and $^{206}\text{Pb}/^{204}\text{Pb}$ source as the CVL. I would be more convinced if the authors showed the compositions of these other WAPM-IB lavas in $^{207}\text{Pb}/^{204}\text{Pb}$ vs $^{206}\text{Pb}/^{204}\text{Pb}$ and $^{208}\text{Pb}/^{204}\text{Pb}$ vs $^{206}\text{Pb}/^{204}\text{Pb}$ space. I don't understand why they were not. Madeira and Cape Verde have compositions that overlap with, but also extend away from those of the CLV lavas. Thus I don't think it is possible to reject a plume hypothesis for Madeira and Cape Verde. In fact, it may be easier to explain these features in terms of a plume mixing with material from the SCLM. Regarding all of the above WAPM-IB lavas (Canaries included), the relatively young volcanism on old lithosphere far from any active makes a non-plume origin all the more difficult to explain.

Reply: The figures of WAPM-IB lavas in $^{207}\text{Pb}/^{204}\text{Pb}$ vs $^{206}\text{Pb}/^{204}\text{Pb}$ and $^{208}\text{Pb}/^{204}\text{Pb}$ vs $^{206}\text{Pb}/^{204}\text{Pb}$ space are inserted as a new Fig. 3. For the Madeira and northern Islands of Cape Verde, the possibility of the involvement of external mantle material are described in the third paragraph of discussion section.

Line 13: This wording is awkwardly, as if you are describing passive continental margins in general. It would be much better rephrased as "Parts of the western continental margin of Africa...."

Reply: We agree. This sentence was revised and expressed as "along the section of the West African passive margin".

Line18: This could be an important argument but is not emphasized in the paper.

Reply: This sentence was revised as "the highly alkaline affinity".

Line 26: This is not true for Cape Verde and Madeira

Reply: Because we have to reduce the abstract, this sentence was removed in the revised manuscript.

Line 32: Canary's have low ^{34}S

Reply: This sentence explains general character of plume signature why it has been explained by whole mantle convection. Thus we did not revise this sentence.

Line 45: Its important to explain to the non-specialist the significance of these compositions.

Reply: Revised. To explain the significance, “suggesting low extents of melting in the same localities for long periods” was added.

Line 49: and possibly low-extents of melting

Reply: Revised as commented above.

Line 77-78: The wording here is unclear. I think you must mean that the Type 2 Bioko samples follow a trend in Pb isotope space shifted to lower 206/204 values compared to Bioko Type 1 samples.

Reply: Revised as “Type 2 Bioko samples follow a trend in Pb isotope space to lower $^{206}\text{Pb}/^{204}\text{Pb}$ values compared to Type 1 samples from the same location”.

Line 78-79: In Supp Fig 4, the open symbols (Type 2) extend to higher (not lower) 143/144 for a give 87/86 compared to the red/orange Type 1 lavas. The extreme Type 2 samples extend to higher 87/86 and lower 176/177 compared to the most extreme Type 1 samples.

Reply: Type 2 samples only exist in SW lavas. Thus, we wanted to compare Type 1 and Type 2 samples only for SW lavas. To make it clear, we added “compared to the Type 1 SW samples”.

Line 80: may (this os a simple explanation but not the ONLY explanation)

Reply: Revised

Line 84: are

Reply: Revised

Line 85: Supplementary Figure 3

Reply: Revised.

Line 85: present-day Atlantic

Reply: Revised

Line 87-88: what is meant by "lowermost" and "spread"? The discussion of the distinction from MORB to this idea is a big jump of ideas. I think you must mean that the geochemical evidence for mixing with SCLM AND its proximity to the craton suggest that cratonic material is part of CVL

Reply: To explain it more clear, one sentence “The isotopically enriched characteristics for the CVL magmas have mainly been explained by the involvement of metasomatized lithospheric mantle component.” was added. We also deleted “lowermost” and “spread” to avoid the confusion.

Line 91: which evaluation? Do you mean the inferences made in the previous sentences?

Reply. To avoid the confusion, this sentence was deleted and more detail explanation was added for each isotope systematics why only Pb isotope can be used.

Line 94: I am not very familiar with this region or with kimberlites. By "present-day" are you implying that you must calculate how the isotope compositions change with radioactive decay and time? It would help me if this were clarified in the text here as well as in Methods.

Reply. Because erupted kimberlite can have variable U/Pb and Th/Pb from the source mantle materials, we have first calculated the initial isotopic compositions of the compiled kimberlite at the eruption age. Then, the possible ranges of isotopic compositions of the current refertilized SCLM were calculated using the plausible U/Pb and Th/Pb ratios as explained in the initial and revised manuscripts.

Line 94-99: Are these kimberlites the basis for defining the isotopic composition of SCLM in Fig. 3? If so, then this should be explained before you present the interpretations you do in the sentences preceding these

Reply: The validity for using the kimberlite as a representative composition for the refertilized SCLM was reported in the cited references. Because detailed explanation how to calculate the present kimberlite-source mantle was written in the METHODS, it made the reviewer confused. In this revised manuscript, thus, the explanation which was written in the Method section was merged with the main text.

Line 101-102: I read the paragraphs describing this calculation in the methods section page 12 and 13. However, I still do not understand how and why lead isotope composition changes with F. Second, the purple lines in figure 1 appears to be displaced relative to the blue type I SW samples. Therefore, I do not understand why it is said that the calculations explain the samples.

Reply: Parent-daughter ratio (U/Pb and Th/Pb) of the asthenospheric mantle-derived melt depends on the degree of melting (F). After the age progression, the isotopic ratio with different F can be different values. We did not explain the isotopic compositions of Type 1 SW as this process. To explain the outline of the calculation, one following sentence was added: "The isotopic compositions of the ancient asthenospheric mantle-derived melt that metasomatized the lithospheric mantle (Fig. 5) were calculated based on the elemental and isotopic compositions of the present MORB-source mantle with various degrees of melting at various age range."

Line 105: It is important to explain what this means.

Reply: To avoid the confusion, "rejuvenated" was deleted and used "Mesozoic E-DMM melt".

Line 106: Above you describe two calculations: one for the present day composition of the kimberlites, and the other for the melting of group 1 kimberlites. I don't understand which model calculation you are referring to here. It is important to state in the text what process these models are simulating.

Reply: To make it clear, the text was revised as "The result (Fig. 5 and Supplementary Fig. 2)..."

Line 113-114: I don't see this at all. In Fig 1, I see ALL of the Type 2 compositions falling within the lower 207/204 and lower 208/204 boundaries of the green outline. There are two samples (open triangles) that extend to or slightly ABOVE the green boundary in 207/204 and 208/204.

Reply: The text was revised by inserting the following detailed explanation which was written in the METHODS section:

"Isotopic data were compiled for mantle xenoliths from cratonic and near-cratonic regions of Africa. Only the pyroxenite vein/layer in the SCLM peridotite of the Congo–Tanzania craton³⁵, which formed ~1.9 Ga and was collected from the Toro-Ankole volcanic region, SW Uganda³⁵ (Fig. 1) have a Sr, Nd, and Pb isotopic composition (no Hf isotopic data were reported) that can be explained by mixing Type 1 and Type 2 magmas (Fig. 5). Thus, the pyroxenite vein/layer in the cratonic SCLM, which formed during the Archaean or Proterozoic eon, is the best candidate for the Type 2 component."

Line 116: what is the evidence for this claim? I see no obvious connection between the shown lava compositions and the yellow pyroxenite field in Fig. 1

Reply: Text was revised as mentioned above.

Line 117: are you referring to the correct figure here? This is a map of seismic tomography

Reply: The location of pyroxenite xenolith (Toro-Ankole) is shown in Fig. 1 (previously Supplement Fig. 1). However, since it may confuse to the reader, we deleted it.

Line 126: originate could implies that the lavas are entirely from the SCLM. It would be better to write "could have been influenced by or partly derived by the same SCLM source(s) as the CVL"

Reply: The geochemical data reveals that the major contribution of these parental magmas were SCLM. Thus, we did not revise the sentence. To make it clear, however, “major source of” was added.

Line 126: as

Reply: revised

Line 136: Given that the WAPM-IB compositions show influence from other sources, that are clearly distinct from the SCLM, how can you refute the possibility of plumes delivering these distinct sources? Do you have a model not involving plumes, and if so, how do you explain these distinct source materials?

Reply: We explained that the variation of isotopic data of WAPM-IB except for Madeira and northern Islands of Cape Verde can be explained by the SCLM components. Thus, we discussed that it is not necessary to consider the involvement of plume materials.

Line 143-144: It seems it is important to show the compositions of the other WAPM-IB lavas in 207/204 vs 206/204 and 208/204 vs 206/204 space.

Reply: The new figure (Fig. 3) was added in the revised manuscript.

Line 146-148: Are the Sr, Nd, and Hf isotopic compositions also similar to Atlas?

Reply: The new figure (Fig. 3) was added in the revised manuscript. No Hf data is available for Atlas samples.

Line 158: I agree for CVL and Canary Islands, Cape Verde is pretty far from the West African Craton

Reply: text was revised as “...and part of the Cape Verde are above the the detached buoyant blob of SCLM stranded in the opening of the ocean basin¹⁶.”

Line 160: Delete “current location of the”

Reply: Revised.

Line 161: were

Reply: Revised.

Line 164: are

Reply: Revised.

Line 165: Contradiction: in the previous phrase you stated they were within the range of Atlantic MORB, how you are stating they are distinct from (Atlantic) MORB.

Reply: Text was revised as “Although the isotopic data for Madeira partially overlap by the range of Atlantic MORB, most of them show distinct compositions relative to the MORB and CVL (Fig. 3).”

Line 166: this seems to come out of nowhere. How do you reach this conclusion?

Reply: Text was revised as “the parental magma in Madeira and northern islands of Cape Verde may contain either a larger proportion of the asthenospheric mantle components than the other WAPM-IB does or an external component not defined by CVL”

Line 167-168: It would help to identify the compositions of the northern islands in Fig. 3. Also, there are many compositions that are clearly trend away from DMM in 143/144 vs 206/204 or are clearly lie off the trend to DMM in PC2 vs PC1 space.

Reply: We tried to use the different symbol for northern Cape Verde samples, but it was not classified because the figure is getting more complicated. Because the location-dependent distinct isotopic

differences for Cape Verde basalts have already reported by several papers, we just cited the reference here. The text was revised as not to identify the other components which was not defined by CVL.

Line 169-170: I do not understand what you are writing here. Are you suggesting that the Cape Verde and Madaera compositions trend away from the SCLM because they are NOT influenced by SCLM during the rifting process, whereas the other WAPM-IB lavas were more heavily influenced by SCLM due to their proximity to rifts?

Reply: The text was revised as “Thus, the parent magmas of Madeira and the northern Cape Verde Island lavas might have originated via different processes from those of the other WAPM-IB. Therefore, it is likely that the degree of involvement of the SCLM in the WAPM-IB source is closely related to the location of the Mesozoic rift axis during the opening of the Atlantic Ocean (Fig. 7).

Line 182-185: low degrees of melting are also consistent with more recent lavas being generated beneath old and thick lithosphere

Reply: This sentence was deleted. But the final paragraph was significantly revised to explain this more in detail.

Line 186-187: you dont know the direction of mantle flow. If its flowing away from the craton then there is no collision.

Revised: The current plate motion is added in Fig. 1.

Line 189: I agree that the alignment of the CLV with the NW edge of the Congo craton as imaged in supplementary Fig 1 long with the lack of age progressive volcanism suggests that lithospheric influence may be important to the formation of the CVL. But the arguments against a plume influence for Cape Verde, Canary, and Madeira are weak in my opinion.

Reply: For the Canary and southern Islands of Cape Verde, we presented that the isotopic data of these lavas can explain without involving the external components (e.g. plume component), which is consistent with the chronological and geophysical data. For the Madeira and northern Islands of Cape Verde, the possibility for the involvement of external materials are described.

Figure 1: The compositions of Canary, Atlas, Cape Verde, and Madeira should be shown in 208/204 vs 206/204 space as well.

Reply: New figures (Fig. 3) were added in the revised manuscript.

Line 212 (Fig. 1): these are model compositions, right?

Reply: Yes. The sentence was revised as adding “Calculated...”.

Line 213 (Fig.1): ??

Reply: revised.

Line 215 (Fig. 1): This is confusing

Reply: The term was changed.

Figure 2: What do the dark blue, light blue, and red colors represent?

Reply: They were indicated in the original figure. However, we re-draw the figure to make it more clear.

Figure 3: It is important to show the compositions of the SCLM derived from the kimberlites in this diagram. & **Line 246 (Fig. 3):** I do not see a light green area.

Reply: The isotopic composition of refertilized SCLM was already shown in the original figure. We re-draw the figure to make it more clear.

Line 299: I dont understand what is being calculated. Measured isotope compositions should be the present-day compositions, no? Are the initial compositions calculated by back-tracking radio-active decay?

Reply: The text was revised as we have replied previously.

Line 323: I'm sorry, but I'm not following what process is causing the isotopic compositions to change with F.

Reply: The text was revised as we have replied previously.

Supplement Fig1: It is imperative that you explain what the colors represent. A colorbar would be best. It appears that red-blue is fast-slow. Are the velocities V_p or V_s ?

Reply: Revised. Added “ V_s models” and definition of colour scales in the figure caption.

The revisions add improvement in some aspects but introduce problems in others. I continue to find the writing problematic. In several places, the writing is verbose, redundant, unclear, inaccurate or seemingly contradictory to what is shown in the figures. Also there are grammatical issues and typographical errors. However, overall I understand the study better and in this sense the improvement is positive. The main arguments for SCLM as the primary source of the CVL appears to be the trace-element characteristics (which are explained by interaction of carbonatite melt and garnet), the shared origin with the kimberlite source material, their distinction from DMM, the seismic evidence for the CVL aligning with the edge of SCLM, and the lack of age progression along the CVL. Then in extending the isotopic evidence to the other WAPM-IB, the authors have successfully argued that compared to the CVL, there is no need for a distinct (e.g., plume) source for the Cape Verde, Madeira, Canary, and Atlas Mt's. The main difficulty I have with a non-plume origin for the Madeira and Canary provinces is the evidence of geographically age progressive volcanism (see figure below). Also, I agree with Reviewer 1 that invoking only edge driven convection (for the provinces without age progression) is not without weakness. Another potential weakness is the volume of volcanos inferred to be produced by only fertile mantle with minimal excess asthenospheric temperature. The strength of the paper is in the geochemical evidence. The proposed solution is not simple, but the value of the paper is in turning the community away from the plume-only explanation and this can motivate further research on such explanations.

Below discuss some issues & make suggestions. I have also made extensive comments directly on the pdf.

I) Line 49 The authors appear to be generalizing too much about the lack of age progressions in the WAPM-IB. I agree that the CVL and Cape Verde's clearly show no geographic age progression (not sure about Atlas Mts but a reference should be given). However, Geldmacher et al. EPSL [2005] argue for age progression along the Canary and Madeira chains, broadly consistent with stationary sources (e.g, plumes or hotspots) beneath a slow-moving African plate (see their figure to the right). Line 49 should be revised and this point should be carefully considered in the discussion near the end of the manuscript.

(II) Line 94-95 and line 145. I do not see how the authors can conclude that the trace-element pattern of Type 1 lavas is similar to Group 1 kimberlites but not Group 2 kimberlites. As shown by my overlays below, all the kimberlites shown show substantial overlap with the CVL compositions.

II) Lines 126-151 The authors have done a better job in explaining how they compute the Pb isotope composition of the kimberlite source but the description is still hard for me to follow and is incomplete.

First, the abstract describes the source of the CVL as SCLM that was metasomatized by kimberlitic melts. My understanding of the use of the word "metasomatized" is that it implies interaction with melts after they were generated from a source in the mantle below where the metasomatism took place. However, the description on p. 4 and 5 seems to be describing the mantle source of the kimberlitic melt. The writing should be adjusted to make it clear whether you are describing the source or the metasomatized region.

Next, regarding the description of the modeling. What I understand is that the kimberlites melts were generated some time ago. Their present-day isotope ratios, however, do not represent the ratios in the mantle source because the distinct U/Pb and Th/Pb ratios between kimberlite and source will mean different rates of radiogenic lead in-growth since the time the kimberlites formed. Therefore to estimate the present-day isotopic composition of the source, one must

- (1) estimate the amount of radiogenic Pb ingrowth in the kimberlites from their U/Pb and Th/Pb ratios
- (2) Subtract that ingrowth to estimate the starting isotope compositions when the kimberlites formed
- (3) Estimate the U/Pb and Th/Pb compositions of the source
- (4) From the starting kimberlite isotope compositions, compute ingrowth in the mantle source to present-day.

Is that correct? If so, then it would help the broad Nature scientific audience if you spelled out the procedure similarly to the above.

Third, to help the reader evaluate the uncertainty of the modeling, I would be good to give the ages of the kimberlites so the reader can get a sense for the amount of ingrowth. It might also be good to show the field of kimberlite Pb isotope compositions in Fig 5 so the reader can have a sense for how the modelled source compositions differ from the data feeding the models. This would help the reader assess the potential uncertainty in the model calculations.

The description of the way in which the present-day Pb isotope composition of the asthenosphere-derived melt was better but could be slightly improved as per recommended revisions above.

(III) Lines 168-191 The discussion of the isotopic source for Type 2 lavas is confusing, inaccurate, or inconsistent with what is shown in the figures. See my comments on manuscript pdf. More importantly, given that most of the Type 2 lavas lie within the green Pb isotope field of the computed kimberlite source, and given that the kimberlite source is subject to (unquantified) uncertainties in the modeling (Fig. 5) I find it inappropriate to argue the Type 2 lavas are derived by mixing with a source that is distinct from the kimberlite source on the basis of Pb isotopes.

(IV) Line 272-300, This discussion of the origin of the WAPM-IB is verbose, not clear in places, and is a bit limited conceptual scope.

First, it is a bit incomplete to discuss edge-driven convection as a source of heat. As proposed by King and Anderson [1998], edge-driven convection causes asthenosphere of normal (not anomalously warm) temperature to rise in places of thin lithosphere and sinking against zones of thick lithosphere (such as beneath the CVL). Melting can occur by decompression (in upwelling) or by heating of fertile material imbedded in SCLM. This insight is not conveyed in the writing.

My recommendation is to discuss the possibility of small-scale convection (SSC) in a more general sense and not be so committed to solely the edge-driven form of SSC. I agree edge-driven convection should receive plenty of discussion given the proximity of CVL to the edge of the high-seismic body. But small-scale convection can also occur independent of thick continental lithospheric edges, due to smaller-amplitude variations in lithospheric thicknesses or density heterogeneity in the asthenosphere, perhaps by SCLM blobs or even the presence of melt [e.g., Raddick and Parmentier [JGR 2002]. See also Ballmer et al. [GRL, 2007; Gcubed 2009]. This is another possibility that could explain some of the WAPM-IB provinces.

It would strengthen the paper if the last section could convey that (again) the proposed solution is not simple (e.g., age progression at Canaries and Madeira) and as noted by Reviewer 1, but the value of the paper is to turn the community away from the plume-only explanation and toward lithospheric and/or upper-mantle processes.

Reply to reviewer's comments:

The revisions add improvement in some aspects but introduce problems in others. I continue to find the writing problematic. In several places, the writing is verbose, redundant, unclear, inaccurate or seemingly contradictory to what is shown in the figures. Also there are grammatical issues and typographical errors. However, overall I understand the study better and in this sense the improvement is positive. The main arguments for SCLM as the primary source of the CVL appears to the trace-element characteristics (which are explained by interaction of carbonatite melt and garnet), the shared origin with the kimberlite source material, their distinction from DMM, the seismic evidence for the CVL aligning with the edge of SCLM, and the lack of age progression along the CVL. Then in extending the isotopic evidence to the other WAPM-IB, the authors have successfully argued that compared to the CVL, there is no need for a distinct (e.g., plume) source for the Cape Verde, Madeira, Canary, and Atlas Mt's. The main difficulty I have with a non-plume origin for the Madeira and Canary provinces is the evidence of geographically age progressive volcanism (see figure below). Also, I agree with Reviewer 1 that invoking only edge driven convection (for the provinces without age progression) is not without weakness. Another potential weakness is the volume of volcanos inferred to be produced by only fertile mantle with minimal excess asthenospheric temperature. The strength of the paper is in the geochemical evidence. The proposed solution is not simple, but the value of the paper is in turning the community away from the plume-only explanation and this can motivate further research on such explanations.

Reply: Thank you for your kind and helpful comments. In this revision, we reconstructed the main text to minimize the verbose and redundant descriptions basically in accordance with the reviewer's suggestions. In addition, new descriptions were added to make the explanation clearer. For this purpose the following major revisions were performed: (1) Supplementary Figure 1 and its caption were moved to Figure 4 and result section, respectively, (2) the calculation method for the isotopic compositions of kimberlite-source mantle compositions were performed not only for Pb isotope but also for Sr, Nd, and Hf isotopes. In this revision, the calculation method was also changed to make the model simpler. (3) By the revision of (2), it was necessary to include the DMM-derived melt component for the Type 1 SW CVL magmas, which could overcome the volume problem. Reply for "the evidence of geographical age progressive volcanism" is shown in the next reply. The edge-driven convection model was revised to the small-scale convection by following the reviewer's comment.

Below discuss some issues & make suggestions. I have also made extensive comments directly on the pdf.

l) Line 49 The authors appear to generalizing too much about the lack of age progressions in the WAPM-IB. I agree that the CVL and Cape Verde's clearly show no geographic age progression (not sure about Atlas Mts but a reference should be given). However, Geldmacher et al. EPSL [2005] argue for age progression along the Canary and Madeira chains, broadly consistent with stationary sources (e.g, plumes or hotspots) beneath a slow-moving African plate (see their figure to the right). Line 49 should be revised and this point should be carefully considered in the discussion near the end of the manuscript.

Reply (L46-47): Although Geldmacher et al. (2005) and other previous studies indicated the age progression along the chain, later studies using more abundant datasets (e.g. Merle et al., 2009; Merle et al., 2018 Australian J. Earth Sci., 65, 591-605; van den Bogaard, 2013) proved and discussed no clear age progression in Canary and Madeira archipelagos. Therefore, it is not necessary to discuss in detail against the Geldmacher et al. (2005) here. Here, the necessary references were cited for each location including Atlas Mountains.

(II) Line 94-95 and line 145. I do not see how the authors can conclude that the trace-element pattern of Type 1 lavas is similar to Group 1 kimberlites but not Group 2 kimberlites. As shown by my overlays below, all the kimberlites shown show substantial overlap with the CVL compositions.

Reply (L94-96): One following sentence: “On the contrary, the trace element patterns of Group 2 kimberlites show enrichments of U and Ba relative to Nb and no negative Pb anomaly, this is distinct from the trace element patterns of the CVL lavas studied here.” was added. Line 145 was also revised.

II) Lines 126-151 The authors have done a better job in explaining how they compute the Pb isotope composition of the kimberlite source but the description is still hard for me to follow and is incomplete.

First, the abstract describes the source of the CVL as SCLM that was metasomatized by kimberlitic melts. My understanding of the use of the word “metasomatized” is that it implies interaction with melts after they were generated from a source in the mantle below where the metasomatism took place. However, the description on p. 4 and 5 seems to be describing the mantle source of the kimberlitic melt. The writing should be adjusted to make it clear whether you are describing the source or the metasomatized region. Next, regarding the description of the modeling. What I understand is that the kimberlites melts were generated some time ago. Their present-day isotope ratios, however, do not represent the ratios in the mantle source because the distinct U/Pb and Th/Pb ratios between kimberlite and source will mean different rates of radiogenic lead in-growth since the time the kimberlites formed. Therefore to estimate the present-day isotopic composition of the source, one must

- (1) estimate the amount of radiogenic Pb ingrowth in the kimberlites from their U/Pb and Th/Pb ratios
- (2) Subtract that ingrowth to estimate the starting isotope compositions when the kimberlites formed
- (3) Estimate the U/Pb and Th/Pb compositions of the source
- (4) From the starting kimberlite isotope compositions, compute ingrowth in the mantle source to present-day.

Is that correct? If so, then it would help the broad Nature scientific audience if you spelled out the procedure similarly to the above.

Third, to help the reader evaluate the uncertainty of the modeling, I would be good to give the ages of the kimberlites so the reader can get a sense for the amount of ingrowth. It might also be good to show the field of kimberlite Pb isotope compositions in Fig 5 so the reader can have a sense for how the modelled source compositions differ from the data feeding the models. This would help the reader assess the potential uncertainty in the model calculations.

The description of the way in which the present-day Pb isotope composition of the asthenosphere-derived melt was better but could be slightly improved as per recommended revisions above.

Reply: This and the related paragraphs and figure were completely revised as follows:

(1) (L138-146): To explain the possible distribution of Group 1 kimberlite-source mantle along the edge of cratons, the following sentences were added: *“The refertilized SCLM represented by the low-Vs cratonic roots could be the most fusible portion of the SCLM and is the most likely candidate for the reservoir of Type 1 magmas (Fig. 5a). Among the available samples, kimberlite has the potential to reveal the nature and composition of the deepest parts of the refertilized SCLM. The location of kimberlites in Africa mostly cluster above the low-Vs zones of the cratonic roots at or near the cratonic margins. Chronological and geochemical data of kimberlites in Africa revealed that the isotopic compositions of Group 1 kimberlite could represent the low-Vs metasomatized zones of the SCLM. The similar trace element pattern between Type 1 lavas and Group 1 kimberlite supports the idea of their genetic linkage.”*

(2) (L151-164): The estimation of the isotopic compositions of the Group 1 kimberlite-source mantle was performed not only for Pb isotopes, but also for Sr, Nd, and Hf isotopes. In the previous calculation method, the ambiguity of the parent/daughter element ratio for the source mantle could make the large uncertainties especially for the Sr, Nd, and Hf isotopes because parent/daughter element had to be given. To omit this unknown factor, the parent/daughter ratios of the source mantle was directly calculated using the non-modal batch melting equation using those of the kimberlite compositions in the revised manuscript.

(3) The isotopic ranges for the kimberlite-source mantle shown in Figure 5 were revised.

(4) (L164-168): The calculated result for the Pb isotopic composition of the source mantle did not change significantly from the previous manuscript. However, Sr, Nd, and Hf isotopes revealed that the Type 1 SW samples cannot have been formed only from the kimberlite-source mantle and necessity for the involvement of DMM components. Thus, the interpretation shown in the revised manuscript changed from the previous one.

(6) Because the length of the abstract had to be shortened within 150 words, i.e. nearly 2/3 from the previous manuscript, the description of “kimberlite melt” was deleted from the abstract.

(7) The field of present day kimberlite composition and the range of age was described in the Figure 5.

(III) Lines 168-191 The discussion of the isotopic source for Type 2 lavas is confusing, inaccurate, or inconsistent with what is shown in the figures. See my comments on manuscript pdf. More importantly, given that most of the Type 2 lavas lie within the green Pb isotope field of the computed kimberlite source, and given that the kimberlite source is subject to (unquantified) uncertainties in the

modeling (Fig. 5) I find it inappropriate to argue the Type 2 lavas are derived by mixing with a source that is distinct from the kimberlite source on the basis of Pb isotopes.

Reply: (L183-206): This paragraph was revised to make it clear that the Type 2 lavas were result of mixing. The assumed end component was denoted as Type 2 enriched end-member components (Type 2 EC). To show the continuous trend from the Type 1 SW to Type 2 samples, trace element ratios were also used for the explanation.

(IV) Line 272-300, This discussion of the origin of the WAPM-IB is verbose, not clear in places, and is a bit limited conceptual scope.

First, it is a bit incomplete to discuss edge-driven convection as a source of heat. As proposed by King and Anderson [1998], edge-driven convection causes asthenosphere of normal (not anomalously warm) temperature to rise in places of thin lithosphere and sinking against zones of thick lithosphere (such as beneath the CVL). Melting can occur by decompression (in upwelling) or by heating of fertile material imbedded in SCLM. This insight is not conveyed in the writing.

My recommendation is to discuss the possibility of small-scale convection (SSC) in a more general sense and not be so commit to solely the edge-driven form of SSC. I agree edge-driven convection should receive plenty of discussion given the proximity of CVL to the edge of the high-seismic body. But small scale convection can also occur independent of thick continental lithospheric edges, due to smaller amplitude variations in lithospheric thicknesses or density heterogeneity in the asthenosphere, perhaps by SCLM blobs or even the presence of melt [e.g., Raddick and Parmentier [JGR 2002]. See also Ballmer et al. [GRL, 2007; Gcubed 2009]. This is another possibility that could explain some of the WAPM-IB provinces.

It would strengthen the paper if the last section could convey that (again) the proposed solution is not simple (e.g., age progression at Canaries and Madeira) and as noted by Reviewer 1, but the value of the paper is to turn the community away from the plume-only explanation and toward lithospheric and/or upper-mantle processes.

Reply: (L311-325): We have completely revised these paragraphs by following the reviewer's comment. We used small scale convection instead of edge-driven convection. Also, the revised manuscript cited the suggested references.

L23: The paper describes the source of kimberlite melts, not mantle that was metasomatized by kimberlites. See comment in my review write-up.

Reply: As we have already replied, this sentence was deleted.

L24: Wrong words. I dont know what a "major location" is

L24: Poor word choice: The locations are located?

Reply (L20): This sentence was revised.

L26: "thermal anomaly" is typically used to describe an area of mantle that is unusually hot, or hotter than most areas of the mantle. Small-scale convection can move mantle of normal temperature to shallower depths or cooler sublithospheric mantle to greater depths.

L27: The paper emphasizes "edge-driven convection" which is only one form of "small-scale convection". If you mean "edge-driven convection" then use those words exactly.

Reply (L22-24): Revised as *"The melting of the source region can possibly be attributed to small-scale mantle convection at the base of the SCLM without the involvement of a mantle plume."*

L38: I think of a "geophysical study" as one that uses geophysical data (seismic, gravity, magnetics, EM). Whereas I would consider the references you cite here as emphasizing geodynamics more than geophysics. That said, I dont see a good reason to name the discipline of these studies.

Reply (L34): "geophysical" was replaced to "geodynamical".

L45: It would be helpful to show a map of these volcanic provinces with ages labelled at the appropriate locations.

Reply: The range of chronological data were labelled in Fig. 1

L49: This is true for Cape Verde, CVL (and maybe the Atlas mountains, but I dont recall), but NOT for the Canaries and Madeira. Geldmacher et al. EPSL [2005] argue for age progression along the Canary and Madeira chains, broadly consistent with stationary sources (e.g, plumes or hotspots) beneath a slow-moving African plate.

Reply (L46-47): Already replied

Line 51: this has not yet been introduced, therefore you had better explain, generally what is meant by this.

Reply: This sentence was omitted by following the reviewer's suggestion.

L52-53: NOt sure why you want to discuss a process that you dont emphasize later in the paper.

Reply: Deleted

L52: This paragraph seems rather verbose and will ultimately be redundant with later parts of the text. It also poses a solution "small-scale convection" before your new evidence is presented, which risks diminishing the value of your new geochemical evidence.

Thus I suggest that this paragraph should only point out the difficulties with the plume hypothesis and then cite the prior work that points to subcontinental lithosphere as being important to the origin (in a general sense): i.e., establish the seismic evidence for lithospheric material beneath many locations of the WAPM-IBs, and other geochemical studies that have called upon SCLM as source material. This tone of approach would set the stage for the study without giving away the punch line.

Reply (L48-58): This paragraph was shortened and several sentences were moved to the discussion.

L54: See my written comments about associating edge-driven convection ONLY with heat.

Reply: This sentence was deleted.

L55: explain that this is images seismically

Reply: This sentence was deleted.

L65-67: This belongs with the previous paragraph.

Reply (L59): This sentence was starts to explain the CVL. Thus we did not connect the paragraph.

L69: Somewhere, such as in this paragraph it would be helpful to introduce that you will be comparing CVL lavas with kimberlites. Otherwise the appearance of "kimberlites" on line 95 seems to come out of nowhere and without any context. The comparison of kimberlites to the CVL lavas is central to the main conclusions of the manuscript. When introducing the kimberlites you might give a short background of them (e.g., of where they are located (mark them in a figure), what their ages are, and why you chose the kimberlites that you worked with.

Reply (L68): We tried to explain the short background of kimberlite here. However, it is explained here, this sentence gets verbose and redundant. Instead, new sentences that explained the high possibility of kimberlite source mantle beneath the cratonic margins were added before calculating the isotopic compositions of kimberlite source materials in the following section (L140-146).

The location and other information is added in the Fig. 1 and Figure 3.

L89-91: The reasoning described here is a bit backwards. Instead, you infer that the negative anomalies originate in the garnet-peridotite stability field because the compositions are well-explained by the partition coefficients between carbonatite melt and garnet.

Reply (L87-89): these sentences were merged and revised.

L93: "could"? What evidence is there for "should"? Can this last phrase be deleted?

Reply (L90) : "should" has been replaced by "could".

L 94-95: First, the inconsistency between the legend and lines in Fig. 2f make it hard for me to distinguish Group 1 and 2, but regardless, ALL of the kimberlite compositions shown in Fig. f show substantial overlap with the CVL. Thus it is not clear that Group 1 and the high-Mg HDFs match better than Group 2. Either be more specific as to how Group 1 and the high-Mg HDF's better match to the CVL compositions than Group 2 or dont leave Group 2 out of this comparison.

Reply (L92-96): The inconsistency between the legend and lines was revised. The text was revised.

L 94-95: Mark the locations of the kimberlites on Fig. 1

Reply: The location of kimberlites were marked in Fig. 1

L98-101: Rather than emphasizing a distinction between the NE, TR, and SW compositions, would it fit with the story of this paper better and be simpler if you described the Type 1 compositions as lying along a binary mixing line between two source materials? Or at least emphasize the similarity of the Type 1 lavas because the more important distinction (as I understand it) is with the Type 2 lavas.

L 103: I do not understand why you describe the Type 2 Annobon & Sao Tome lavas different than the Type 2 Bioko lavas. Wouldn't it be simpler to describe (all) the Type 2 samples as having higher 207/204 and 208/204 at a given value of 206/204 relative to Type 1.

L104 & 105: do you mean geographic location or for the same value of 206/204?

L105: Are you discussing only the Type 2 Bioko samples or all Type 2 samples?

Also the way the sentence is worded makes it unclear why you are comparing Type 2 lavas with the Type 1 SW lavas. I see in Fig 3c (i.e., 143/14 vs 87/86) that the open squares, diamonds overlap with the Type 1 SW compositions and that the open triangles appear to extend away from the Type 1 SW compositions. Thus it would be more straightforward to explain the Type 2 SW lavas as overlapping the Type 1 SW lavas but extending to higher 87/86 and lower 144/143.

That said the Type 2 Bioko samples appear to overlap more with the Type 1 NE lavas.

Reply (L97-131): This paragraph was revised. Because different between Type 1 NE and Type 1 SW is the most important classification to identify the different source materials, they were first described. In this revision, difference of trace elements among two groups were moved from the caption of supplementary Fig. 2 to the main text. Then, difference between Type 1 and Type 2 was described in the subsequent paragraph. The one-by-one points suggested by the reviewer were revised.

L114-122: There is a far bit of redundance of this paragraph with concepts introduced in lines 49-64 and with the discussion towards the end of the document. Presenting the concept of the involvement of SCLM here and developing the evidence later in the paper does not convey a logical progression of ideas.

This paragraph emphasizes prior geochemical studies that calls on SCLM, which is background that might be better placed in the introductory paragraph on lines 49-64.

L116: This is redundant

Reply (L120-131): These sentences were moved to the introductory section and partly to the discussion.

L137: Do you mean the source of the kimberlite melts or the mantle that was metasomatized by kimberlites (as stated in the abstract)?

Reply (L147): It is kimberlite-source mantle as will be described in L152.

L138: See my comment in the review about explaining what you are doing here.

Reply (L154-): Already replied

L 142: of the starting source material and found that these ratios

L 143: The last step that would help me is to articulate that from U/Pb and Th/U estimates of the source, you simulated radioactive decay over X Myrs to arrive at isotope compositions assuming the isotope compositions of the kimberlite was the composition of the source X Myr ago.

Reply: These sentences were deleted.

L 146: This has NOT been established. See my comment on lines 94-95. Also, the sentence is problematic because it begins by stating that the evidence against Group 2 kimberlites for the source of CVL is the trace-elements, but then the sentence finishes by describing isotopes. Which is the main evidence trace-elements, isotopes, or both?

Reply (148-152): To make clear the meaning, this sentence was revised.

L 157: no need for all caps

Reply (L174): Revised

L 164: Reverse the order of these 2 new sentences.

Reply (L178-182): Revised

L 169: (1) You should reference Fig 5 in this sentence

(2) You need to make it clear that "isotopic trend" is referring to the "isotopic trend of the Type 2 lavas"

(3) There is a typographical error because 208/204 is listed twice.

(3) I do not fully understand or agree with this sentence.

In fig 5, I see open triangle extending to higher (not lower) 208/204 and slightly high (not low) than the green field of the kimberlite source. Thus if anything the data show evidence for source material with higher (not lower) 208/204 and 207/204 at a given 206/204 than the kimberlite source.

That said, because there is so much overlap with the green field and because the green field has (unquantified) uncertainties associated with modeling I think you are too far out on a limb to make the above assertion.

Reply (L183-185): These sentences were revised following the reviewer's comment.

L 170: What do you mean by "This"?

Reply: This sentence was revised and moved to the L124-125.

L171: wrong figure

Reply: Revised

L175: You should make it more clear that the source of EM1 was estimated by ref. 11 (not part of this study)

Reply (L188): To make it clear, "in the previous study" in added in this sentence.

L 179: I think you mean Type 2 are more similar to Group 2 than Group 1 kimberlites

Reply: This sentence was deleted.

L181: The computed Group 2 compositions are shown in Fig. 5 not Fig 3.

Reply: This sentence was deleted

L184: This sentence is either poorly written, or I am not understanding it, or the sentence is wrong. Is the pyroxenite you are referring to within the yellow field of Fig. 5? If not, this composition must be shown.

If it is in the yellow field, then the yellow field lies completely outside the range of compositions of the CVL so no part of the yellow field can be a mixture of Type 1 and Type 2 lavas.

However, I agree that the high 207/204 and 208/204 end of the yellow field starts to approach the Type 2 compositions, and hence could represent the end-member mixing component controlling the Type 2 lavas.

L 187: "the Type 2 component" is a bit vague. "the mixing component controlling the Type 2 compositions" or "the end-member component influencing Type 2" is better.

Reply (L126-131): Revised by following the reviewer's suggestion. In the Figure 3, the direction of Type 2 EC was presented.

L 188: It would have helped me if this were worded something like "Type 2 magmas occurring with Type 1 only on the volcanos of the SW part of the CVL" (Type 2 lavas coexist with all lavas, not just Type 1 SW; the samples are in your lab)

Reply: It wa already described in L81-86.

L189: So why does this co-existence occur ONLY in the SW and not the NE?

Reply (L203-206): One following sentence is added: "This can occur because both source materials (i.e. kimberlite-source mantle and pyroxenite vein/layer) can coexist near the bottom of the refertilized SCLM (Fig. 5a)."

L200: Does this work for 87/86 as well?

Reply (L212-214): It works for 87Sr/86Sr as well. In the revised manuscript, all the analysed isotope systematics were presented in Figure 6.

L220: If the Type 1 NE lavas represent strong influence by HIMU, why do you label a box "Mesozoic E-DMM melt" next to them in Figure 5?

Reply (L236-239): Because Pb isotopic data for Type NE are different from the typical HIMU values, this sentence was revised as *"The distinct Pb isotopic values of these high $^{206}\text{Pb}/^{204}\text{Pb}$ NE CVL and Atlas lavas from the typical HIMU components, represented by St. Helena basalts (Supplementary Fig.2), indicates that the involvement of St. Helena type plume is unlikely for the generation of CVL and Atlas magmas as suggested previously."*

L226: These paragraphs might be better joined. As is, the purpose of the top paragraph is a bit up in the air.

Reply (L233): These two paragraphs were joined.

L238: This is very speculative. It would be safer to explain that it is near a high-velocity blob that could be SCLM.

Reply: This sentence was deleted.

L249: This belongs in the previous paragraph

Reply (L296): This and following sentences were joined in the previous paragraph.

L252: This is a bit redundant with the last sentence of the previous paragraph.

Reply: This sentence was deleted.

L253: Start a new paragraph here.

Reply (L288): We did not start a new paragraph her. However, the preceding sentence was revised.

L256: Again, do you attribute this to HIMU or E-DMM?

Reply (L289): To avoid the confusion, "HIMU" was exchanged to "high 206Pb/204Pb".

L259: You need a topic sentence that better encompasses the purpose of this paragraph, which seems to be the origin or the process by which the Canary and Atlas regions formed

Reply (L263): This sentence has been moved in the third paragraph of discussion.

L263: poor word choice

Revised: This sentence was deleted.

L278: In the next sentence you argue against (high) thermal anomalies. A thermal anomaly could be high and localized, there this sentence could be contradictory with the next.

For this sentence do you mean "minor thermal" anomaly or "low-amplitude thermal" anomaly here?

Reply (L311-312): This sentence was moved to the next paragraph, and revised as "The generation of SCLM parental magmas can thus be triggered by low-amplitude thermal or compositional anomalies."

L281: Grammar problems: A model (of edge driven convection) is not a heat source. Edge-driven convection can supply the heat to form magmas. But heat is not the only ingredient needed for melting, therefore I suggest "Edge-driven convection has been proposed as the mechanism for creating the CVL...."

L281: This paragraph is verbose, grammatically problematic, unclear, or inaccurate.

L302: rising from the lower mantle ("spanning the whole" could be misinterpreted as a plume filling up the mantle around the whole earth).

Reply (L311-325): This paragraph was completely revised by following the reviewer's comment.

L322 (Fig.1): Are these relevant to this study? If not, then explain why they are marked, or delete the white triangles.

Reply: Because white triangles are shown in the original figure of reference (O'Reilly et al., 2009), it is impossible to delete them.

Fig.2: This key does not match the colored lines in the diagram. What are the dark blue and magenta dashed lines?

Reply: Revised

Fig 3: please mark NE, TR, and SW like you do in the legend of Fig. 5

Reply: Revised. The term TR was omitted from the text and figures.

REVIEWERS' COMMENTS:

Reviewer #3 (Remarks to the Author):

Review of revised manuscript by Belay et al. "Non-hypothesis for the genesis of passive continental margin oceanic basalts", by G. Ito

The manuscript is much improved and adequately address the comments I raised in my earlier review. I have a few remaining minor comments, which I have inserted in the PDF document. Perhaps the most significant (and still minor) comment that I have is that the age ranges of the volcanics for Cape Verde (0-17Ma), Canary (0-142Ma), Madeira (0-97Ma), Atlas (0-67 Ma), and CVL (0-31) on Fig. 1 are quite a bit younger than the rift zones as labelled in Fig. 8 (185 Ma, and 120-115 Ma). This would suggest that these volcanic provinces did not perform near the active ridge axes. The discussion on page 9 should therefore be revised accordingly. Beyond this, I look forward to seeing this manuscript published.

Reviewer #3 (Remarks to the Author):

Review of revised manuscript by Belay et al. "Non-hypothesis for the genesis of passive continental margin oceanic basalts", by G. Ito

The manuscript is much improved and adequately address the comments I raised in my earlier review. I have a few remaining minor comments, which I have inserted in the PDF document. Perhaps the most significant (and still minor) comment that I have is that the age ranges of the volcanics for Cape Verde (0-17Ma), Canary (0-142Ma), Madeira (0-97Ma), Atlas (0-67 Ma), and CVL (0-31) on Fig. 1 are quite a bit younger than the rift zones as labelled in Fig. 8 (185 Ma, and 120-115 Ma). This would suggest that these volcanic provinces did not perform near the active ridge axes. The discussion on page 9 should therefore be revised accordingly. Beyond this, I look forward to seeing this manuscript published.

Garrett Ito

Thank you for your kind and very helpful comments.

I might have made the reviewer misunderstand. We did not explain that the magmatism of WAPM-IB was active near the active ridge axis. The locations of the WAPM-IB were far off the active Atlantic ridge axis when the magmatism started. I think that this misunderstanding can be caused by the insufficient explanation for Figure 8. Thus, the additional explanations were added in page 9.

Line 27: there are two transport processes implied here, one from the deep mantle up to OIBs (via mantle plumes) and the other down from the lithosphere during subduction. Describing only one, and not indicating down vs up makes the sentence confusing.

Reply: This sentence was revised, explaining only for the involvement of recycled materials.

Line 33: This word implies you have already introduced the plume hypothesis, but it has not yet been introduced in the current discussion.

Reply: "also" was deleted.

Line 35 The general reader may not understand the link between "intraplate magmatism" and "OIB" as stated in the 1st sentence

Reply: "Intraplate" was changed to "OIB".

Line 37: redundant use of "west"

Reply: Revised.

Line 49: it might be more appropriate to be more specific here. i.e., "in the shallow upper mantle"

Reply: Revised.

Line 54: It was not established that the delaminated lithosphere referred to in the previous sentence is indeed derived from the local (sub) continental lithosphere; therefore, this is too big a jump from the previous sentence.

Reply: This sentence was replaced by "However," .

Line 59: It is arguable whether this is in fact "unique". Both the Cape Verde and the Canary-Atlas chains are near the edge of the high velocity continental lithosphere. The sentence would avoid this potential point of argument by making the deletion I recommend.

Reply: Deleted

Line 92: I do not see a negative anomaly for Sr in Fig. 2

Reply: Deleted.

Line 161: It is good to know how sensitive the results are to this, not well constrained, percentage. One way to do this for example would be to show results for a range of percentages (e.g., 1% and 3%)

Reply: The calculate results between 0.5 and 3 % were shown in the revised figure.

Line 203: I am not following the argument here. What is it about the Southwest location that makes the presence of pyroxenite more likely in the SW vs the NE? In figure 5, the light blue pyroxenite appears to be present everywhere there is re-fertilized SCLM.

Reply: The pyroxenite vein/layer from Toro-Ankole was interpreted as had formed between 0.2 and 1 Ga, possibly be related to kimberlitic melt. Thus, it is likely that the source of Type 2 magmas could coexists with the Group 1 kimberlite source mantle rather than the source region that metasomatized by Mesozoic rift system. These sentenced were revised to explain these arguments.

Line 216: Do you mean 4 principal components? If so, then you should show arrows in the figure indicating the directions of PC 3 and PC 4. Or better yet, if you did a PCA analysis for the other lavas, indicate whether you recover approximately the same for principal components as the CVL?

Reply: As described in the figure caption, the first and second principal components count 98.9 % of the total variability of the Pb isotope variation of WAPM-IB. Then, we explained that the variation of most lavas can be explained the mixing of same source materials that identified from CVL. The position of the estimated kimberlite-source mantle compositions was indicated.

Line 283: The age ranges of the volcanoes shown in figure 1 would put the volcanoes distant in time from the rift zone at 185 Ma (shown in Fig. 8). Depending on the spreading rate and direction, they could be far in space as well. This is especially true if one considers the full range of ages.

Reply: I have made the reviewer misunderstand the meaning of Figure.8. The location of WAPM-IB are illustrated with the current distance from the coast line of African continent. Thus, the position of Mesozoic rift axis roughly indicate the position of the current continent-oceanic boundary. Thus, the location of the OIB does not change by the spreading rate and/or direction.

Line 293: All of them would be far from the active zone of rifting (see comment above)

Reply: Same as above.

Line 303: High (or primordial) $3/4$ is often associated with mantle plumes from the lower mantle, not asthenosphere.

Reply: Revised.

Line 313: SSC is not a "melting process", it is a dynamical process of the upper mantle that can lead to melting.

Reply: Revised.

Line 320: This is not shown in figure 1 so it would be good to identify the depth at which this was identified.

Reply: Because it has already explained as "tabular-shaped low-velocity zones that extend to a depth of ~300 km" in line line 264, it will be redundant to write it again here.

Figure 2: I am having trouble aligning the lines with elements; it would help to label all 3 horizontal axes with the elements.

Reply: The labels with lines were added.

Figure 5: An inconsistency with this figure, is that you show small-scale convective download beneath the volcanoes. Such downloading, presumably would inhibit melting. You might consider the discussion by S. King in <http://www.mantleplumes.org/EDGE.html>

Reply: The flow of small-scale convection cell was revised. The upward flow of small-scale convection was properly illustrated beneath the melting region. The position of upward flow was adjusted below the melting region.